

# DREAM: Drafting with Refined Target Features and Entropy-Adaptive Cross-Attention Fusion for Multimodal Speculative Decoding

**Yunhai Hu**[1]    **Tianhua Xia**[2*]    **Zining Liu**[4*]    **Rahul Raman**[1]    **Xingyu Liu**[2]
**Bo Bao**[3]    **Eric Sather**[3]    **Vithursan Thangarasa**[3]    **Sai Qian Zhang**[1,2]

[1]Courant Institute of Mathematical Sciences, New York University
[2]Tandon School of Engineering, New York University
[3]Cerebras Systems Inc.
[4] University of Pennsylvania

{yh5961, tx856, rr4549, xl5444, sai.zhang}@nyu.edu
zliu0@seas.upenn.edu
{bo.bao, eric.sather, vithu}@cerebras.net

## Abstract

Speculative decoding (SD) has emerged as a powerful method for accelerating autoregressive generation in large language models (LLMs), yet its integration into vision-language models (VLMs) remains underexplored. We introduce *DREAM*, a novel speculative decoding framework tailored for VLMs that combines three key innovations: (1) a cross-attention-based mechanism to inject intermediate features from the target model into the draft model for improved alignment, (2) adaptive intermediate feature selection based on attention entropy to guide efficient draft model training, and (3) visual token compression to reduce draft model latency. DREAM enables efficient, accurate, and parallel multimodal decoding with significant throughput improvement. Experiments across a diverse set of recent popular VLMs, including LLaVA, Pixtral, SmolVLM and Gemma3, demonstrate up to **3.6× speedup** over conventional decoding and significantly outperform prior SD baselines in both inference throughput and speculative draft acceptance length across a broad range of multimodal benchmarks. The code is publicly available at: https://github.com/SAI-Lab-NYU/DREAM.git.

## 1   Introduction

Large language models (LLMs) have shown impressive performance across diverse tasks, but their inference speed remains limited due to the standard autoregressive process, which includes both prefilling and decoding stages. To overcome this limitation, speculative decoding (SD) [48, 5, 19] accelerates the autoregressive process by splitting it into a low-cost drafting stage and a parallel verification stage, enabling multiple drafted tokens to be validated in a single pass through the target LLM. This approach accelerates the decoding stage while preserving the output quality of the target model through a mechanism of acceptance and rejection.

While these techniques have been extensively developed to accelerate inference in text-only LLMs [25, 24, 26, 4, 2, 55, 61, 40, 6, 50, 15], there has been limited work integrating SD into multimodal LMs (MLLM) [20, 43], especially vision-language models (VLM) [10]. VLMs differ from their text-only counterparts by requiring seamless integration of visual and textual information. This integration process typically occurs in two stages: first, extracting meaningful representations from images, and second, applying language reasoning capabilities to generate appropriate responses. For example, in

---

[*]Authors contributed equally; the order of authorship was assigned randomly.

39th Conference on Neural Information Processing Systems (NeurIPS 2025).

LLaVA [29], a vision encoder transforms images using 24 layers of self-attention and feed-forward networks. These features are then projected into the text space and fused with text embeddings within the language model's backbone. The effectiveness of this process heavily depends on the model's ability to maintain coherent representations across modalities, which presents unique challenges for acceleration techniques.

In this paper, we propose _Drafting with Refined Target Features and Entropy-Adaptive Cross-Attention Fusion for Multimodal Speculative Decoding_ (DREAM). Specifically, DREAM employs a specialized cross-attention mechanism that enhances the interaction between visual and textual features, ensuring that key information in the target model is adequately captured even in the draft generation stage. Another key innovation of DREAM lies in its selective use of intermediate-layer representations, which encapsulate the most informative features from both modalities to effectively supervise the draft model with high accuracy. Finally, DREAM introduces a visual input compression scheme for the draft model, guided by the intermediate features from the target model, which substantially reduces processing latency without compromising accuracy.

We evaluate DREAM on a diverse set of popular VLMs, including LLaVA-v1.6-Vicuna-7B/13B [29], SmolVLM-2B [38], Pixtral-12B [1], and Gemma3-12B [3], across multiple multimodal tasks. Our extensive experiments demonstrate that DREAM substantially outperforms well-established SD methods, while consistently achieving high acceptance rates across various multimodal applications, such as table structure recognition, interactive segmentation, animal keypoint detection, and chart question answering. Our main contributions are summarized as follows:

- DREAM incorporates a cross-attention mechanism to leverage intermediate outputs from the target model, facilitating more effective knowledge transfer to the draft model and leading to significant performance gains.

- During DREAM training phase, intermediate features from the middle layers of the target model are dynamically selected based on their entropy and used to supervise the draft model. This enhances the draft model's predictive accuracy and increases token acceptance lengths.

- We investigate the relative importance of visual input and show that visual input in the draft model can be effectively compressed using a scheme guided by intermediate target features, reducing draft processing time and overall speculative decoding latency.

- Evaluation results show that these three strategies enable DREAM to achieve up to a $3.6\times$ reduction in latency compared to conventional decoding methods across a range of VLMs and tasks, outperforming existing speculative decoding approaches.

## 2    Background and Related Work

### 2.1    Speculative Decoding

Speculative decoding (SD) [48] has proven effective in mitigating the sequential bottleneck in language model inference. It operates in two stages: a lightweight draft model rapidly generates a sequence of candidate tokens, which are then verified in parallel by a more accurate target model. Specifically, the draft model generates tokens $d_1, d_2, \ldots, d_\gamma$, where $\gamma$ is a predefined number of tokens. These candidates are verified concurrently by the target model but accepted sequentially. Specifically, a draft token $d_i$ is accepted with probability $\min(1, P_{\text{target}}(d_i)/P_{\text{draft}}(d_i))$, where $P_{\text{draft}}$ and $P_{\text{target}}$ are the probabilities as-

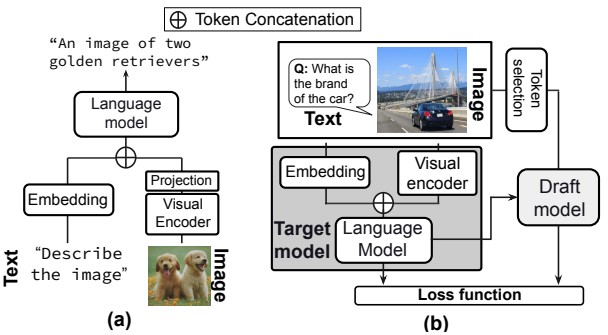

Figure 1: (a) Standard VLM. (b) DREAM overview.

signed by the draft and target models, respectively. Otherwise, the token is rejected. This acceptance rule is compatible with various sampling temperatures. If a rejection occurs at $d_i$, all subsequent

draft tokens from $d_i$ onward are discarded, and the target model's token $t_i$ is used by draft model to continue generation. If all tokens are accepted, the draft model proceeds to generate next batch.

Building on this foundation, researchers have proposed a range of techniques to enhance the efficiency of the draft-verify process. For drafting, various approaches have been developed to eliminate or improve draft models. Medusa [4] introduces lightweight decoding heads on the target model itself, eliminating the need for a separate drafter. Self-speculative techniques include layer skipping [14, 61, 8, 28, 32, 56], which accelerates draft generation by selectively processing fewer transformer layers. Other strategies leverage token distillation [64], N-gram prediction [42, 49, 34], and retrieval-augmented drafting [57, 13, 54] to improve draft quality while minimizing computational overhead. Tree-based verification methods [40, 52, 6, 47, 4] support parallel exploration of multiple completion paths, substantially improving throughput over traditional linear verification. EAGLE [25] introduces feature-based uncertainty estimation for tree construction, while EAGLE-2 [24] further improved this with dynamic context-aware trees. Numerical optimization-inspired approaches include Jacobi iterations [45] and Lookahead decoding [9], which reformulate autoregressive generation as parallel optimization problems. For production deployment, various frameworks enhance system-level efficiency. TRIFORCE [50] employs hierarchical speculative decoding combined with a sparse KV cache to support ultra-long sequences exceeding 100,000 tokens. Parallel scheduling approaches [41, 33] enable draft generation to run concurrently with target verification, while heterogeneous compute solutions like Dovetail [62] optimally distribute models across CPU/GPU resources.

While the aforementioned techniques focus on SD for language-only models, there has been limited exploration of SD in MLLMs. In speech synthesis, VADUSA [20] applies SD to accelerate inference in text-to-speech systems, simultaneously improving synthesis quality. Building on the principles of SD, the authors of [43] propose a multi-token prediction mechanism that significantly boosts inference efficiency for speech generation. In the VLM context, [10] applies speculative decoding to the LLaVA-7B model, demonstrating up to $2.37\times$ speedup by utilizing a lightweight, language-only draft model under memory constraints. IbED [18] proposes an *In-batch Ensemble Drafting* strategy that employs ensemble techniques at the batch level without introducing additional model parameters. In contrast, DREAM introduces a novel cross-attention mechanism and adaptive intermediate feature selection to enhance draft model training, resulting in substantial latency reductions.

## 2.2 Vision Language Model and Computation Profiling

Vision-Language Models (VLMs) are designed to jointly process visual and textual inputs, allowing machines to interpret and generate content that combines both modalities. As illustrated in Figure 1 (a), a typical VLM consists of a visual encoder and a language model. The input image is first processed by the visual encoder to generate visual tokens, which are then concatenated with the textual tokens. These combined tokens are passed to the language model, which generates the final textual outputs. Several VLMs [23, 22] have been developed. Recent works like LLaVA [31], InstructBLIP [7] and Pixtral [1] aim to enhance the zero-shot capabilities of VLMs by better aligning them with human preferences. While the large model sizes have resulted in significant performance improvements, their

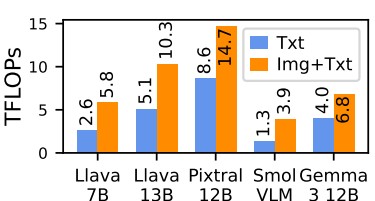

Figure 2: Computational cost of VLMs processing text only (Txt) and multi-modal (Img+Txt) inputs.

computational complexity and storage requirements limit their deployment on resource-constrained devices. Lightweight VLMs, such as TinyGPT-V [59] and TinyLLaVA [63], explore the potential of small-scale models and focus on developing efficient VLM architectures. The recent development of SmolVLM [38] introduces a family of compact VLMs with parameters ranging from 256M to 2B, achieving exceptional performance while maintaining smaller model sizes.

To quantify the computational cost introduced by visual inputs processing in VLMs, we profile the floating point operations (FLOPs) required by various models, including LLaVA-v1.6-Vicuna-7B [31], LLaVA-v1.6-Vicuna-7B, Pixtral-12B [1], SmolVLM-2B [38], and Gemma3-12B [11] over the ScienceQA dataset. We select a typical sample which contains a $480 \times 300$ image, a prompt with 166 tokens, and use the model to generate 500 tokens, representing an average case across the dataset. FLOPs are measured using PyTorch Profiler. Figure 2 compares the TFLOPs for each model when processing text-only (Txt) versus multimodal (Img+Txt) inputs, revealing an average $2.1\times$ increase

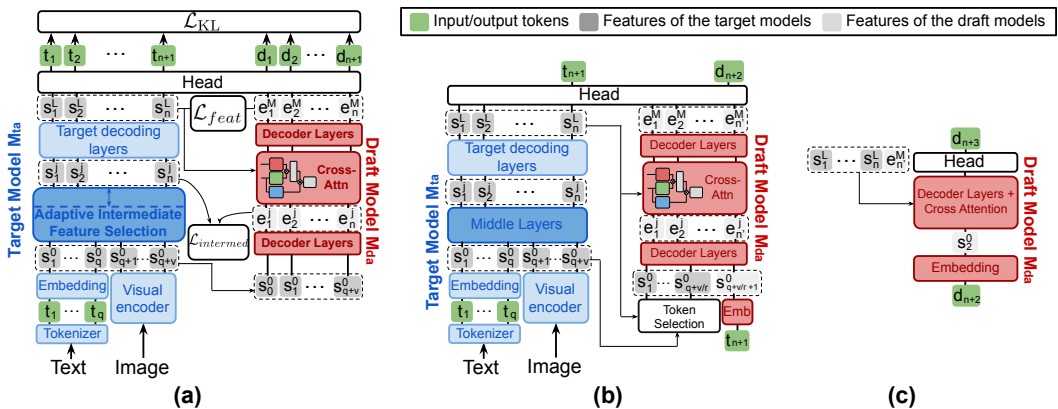

Figure 3: (a) illustrates the training paradigm of DREAM, while (b) and (c) depict its inference workflow. For simplicity, the tree decoding is not shown in (b) and (c).

in computation with the inclusion of visual data and highlighting the need for more efficient visual processing methods.

## 2.3 Intermediate Feature Distillation

Intermediate Feature Distillation (IFD) enhances traditional knowledge distillation by aligning student models not only with the teacher's final outputs but also with its intermediate representations. FitNets [44] introduced the idea of using intermediate *hints*, showing that projecting selected teacher layers through lightweight adapters can improve compact student models. Later methods refine this by identifying high-similarity layers [60], using attention-based weighting [27], or dynamically selecting task-relevant features, as in TED [27]. CoDIR [51] further introduces contrastive losses for tighter feature alignment. In multimodal contexts, OLA-VLM [16] and VLsI [17] adapt these ideas to distill visual embeddings into language representations. Recently, Skean et al. [46] proposed evaluating per-layer representations using information compression, geometric separability, and robustness, showing that mid-layer features often outperform final-layer ones across tasks, revealing a non-monotonic trade-off between information richness and task relevance.

## 3 Methodology

Figure 1 (b) provides an overview of DREAM, where the draft model receives the textual input alongside a subsampled visual input to accelerate output generation. Additionally, the intermediate features are adaptively selected to better guide the training of the draft model. Figure 3 presents the detailed training and inference scheme of DREAM. Specifically, let $M_{ta}$ and $M_{da}$ represent the target and draft models, respectively, with a total of $L$ and $M$ transformer blocks, respectively. let $t_n$ and $d_n$ denote the $n$-th token produced by $M_{ta}$ and $M_{da}$. For $M_{ta}$, assume the textual prompt consists of $q$ tokens and the visual input comprises $v$ tokens, respectively. We use $s_n^{j-1}$ and $e_n^{j-1}$ to denote the $n$-th intermediate feature token at layer $j$ for $M_{ta}$ and $M_{da}$, respectively.

Figure 3 (b) highlights the DREAM inference process. Let $r$ denote the subsampling factor, and $q$ and $v$ represent the number of prompt and visual tokens, respectively. As illustrated in Figure 3 (b), the target model processes all $q + v$ input feature vectors $(s_1^0, \ldots, s_{q+v}^0)$. In contrast, with our subsampling method applied, the draft model processes only $q + \lfloor v/r \rceil$ tokens, substantially lowering the computational load and improving the decoding speed of the draft model, as motivated in Section 2.2. Next, we depict the cross-attention architecture of DREAM (Section 3.1), followed by the adaptive intermediate feature selection strategy for training the draft model (Section 3.2). Finally, we introduce the visual token compression method used to accelerate the draft model in Section 3.3.

## 3.1 Cross-attention Mechanism for Efficient Knowledge Injection

To more effectively transfer knowledge from the target model and enhance the performance of the draft model, we introduce a cross-attention mechanism to fuse features from the target model

to the draft model. This mechanism naturally aligns with the dual-model setup, using the draft's newly generated token embeddings as queries to efficiently retrieve cached target features. This lightweight attention layer integrates long-range, multimodal context into the decoding process. The attention weights act as soft gating, enabling adaptive selection of the refined target visual and textual cues. Crucially, unlike the EAGLE family of methods [25, 24], which concatenate draft token embeddings with precomputed target features and process them jointly, DREAM adopts a more structured fusion strategy. While EAGLE's concatenation approach is effective for purely textual tasks, it tends to weaken structured visual representations by treating visual and textual features as a simple concatenated sequence, potentially disrupting the spatial relationships learned by the vision encoder. In contrast, our cross-attention mechanism adaptively retrieves relevant visual and textual cues, preserving the integrity of multimodal information. This leads to more effective knowledge transfer and improved draft model performance, as demonstrated in our evaluation results in Section 4.2.

As depicted in Figure 3 (b), in the beginning decoding stage, both textual and visual prompt tokens $t_1, \ldots, t_n$ are initially fed into the target model, which then begins generating the next token $t_{n+1}$, where $n = q + v$. The draft model $M_{da}$ subsequently predicts the $(n + 2)$-th token, denoted as $d_{n+2}$. To better help the draft token generation, the cached last-layer features from the target model $M_{ta}$, denoted as $S^L = (s_1^L, s_2^L, \ldots, s_n^L)$, and the intermediate features from the draft model $M_{da}$, denoted as $E^j = (e_1^j, e_2^j, \ldots, e_{n'}^j)$, where $n' = q + \lfloor \frac{v}{r} \rfloor + 1$, are integrated using a cross-attention mechanism. In this setup, $E^j$ is used as the query, while $S^L$ provides the keys and values. With $z$ denoting the dimensionality of queries and keys, the cross-attention is then computed as follows:

$$Q = E^j W_Q, \quad K = S^L W_K, \quad V = S^L W_V, \quad F = \text{softmax}\left(\frac{QK^\top}{\sqrt{z}}\right) V \tag{1}$$

where $W_Q, W_K, W_V$ denote the weight matrices in the draft model. The fused features $F$ replace the original first-layer features and are propagated through the subsequent decoder block. For simplicity, we assume a single attention head in this illustration. After generating the first token $d_{n+2}$, the draft model leverages cross-attention over both the previously verified features from the target model and its own final-layer features. Specifically, we concatenate the target features $S = (s_1^L, s_2^L, \ldots, s_n^L)$ with the latest draft feature $e_{n'}^M$ to form the key and value set $S = (s_1^L, s_2^L, \ldots, s_n^L, e_{n'}^M)$ for generating the next draft token $d_{n+3}$, as described in Figure 3 (c).

Moreover, we adopt the top-$k$ reordering logic and tree-masking strategy introduced in EAGLE-2 [24]. Specifically, we use the newly generated first-layer draft tag features and perform a tree-based width-$k$ expansion to construct the candidate feature set $E = (e_{n'+1}^1, \ldots, e_{n'+k}^1)$. The tree mask is then applied to filter out irrelevant tags, resulting in the final representations $E_{tree} = (e_{n'+1}^M, \ldots, e_{n'+k}^M)$, which are used to simultaneously generate $k$ new draft tokens. This approach enables efficient parallel decoding, and the same tree mask is also leveraged during verification to reduce computation overhead. More details are provided in the supplementary materials on tree.

### 3.2 Adaptive Intermediate Feature Selection for Draft Training

In this section, we detail the training strategy for the draft model. To help the draft model effectively learn the behavior of the target model $M_{ta}$, we utilize intermediate-layer features from the target model as supervision signals. For these features to effectively guide the training of the draft model $M_{da}$, they must satisfy two key criteria: they can provide key information and capture rich semantic content, and they should be essential, exhibiting low variability across tokens, to enable faster and more stable learning in the draft model. As discussed in Section 2.3, intermediate features with low token-level attention entropy are well-suited for distillation, as they focus on salient content and offer stable guidance. DREAM implements a simple yet effective mechanism to select such features from each layer of the target model to guide the efficient training of the draft model, which is shown in Figure 3 (a). For the $l$-th decoder block, its input tokens and output tokens are denoted as $S^{\ell-1} = (s_1^{\ell-1}, s_2^{\ell-1}, \ldots, s_n^{\ell-1})$ and $S^\ell = (s_1^\ell, s_2^\ell, \ldots, s_n^\ell)$, respectively. Let the attention matrix $A_\ell$ associated with $l$-th layer as $A_\ell = \text{softmax}(\frac{Q_\ell K_\ell^\top}{\sqrt{z}})$, where $Q_\ell = S^{\ell-1}W_Q$ and $K_\ell = S^{\ell-1}W_K$, the average attention entropy (AE) is calculated as:

$$\text{AE}(\ell) = -\frac{1}{n}\sum_{i=1}^{n}\sum_{j=1}^{n} A_{\ell,i,j} \log A_{\ell,i,j} \tag{2}$$

where $A_{\ell,i,j}$ denotes the $(i,j)$-th element of $A_\ell$. In practice with multiple heads, AE($\ell$) is also averaged over all attention heads. During each decoding step $n$, we dynamically select the layer $\ell^\star$ with the lowest average entropy, defined as $\ell^* = \arg\min_{\ell \in L} [AE(\ell)]$. We then distill the information from $s_i^{\ell^\star}$ into the initial decoder block of the draft model using a smooth $\ell_1$ loss, guiding the draft model to align with the most informative intermediate representation of the target model. This adaptive feature distillation strategy leads to improved performance by minimizing the $\ell_1$ loss between the feature vector of the $m$-th layer of the draft model $E^m = (e_1^m, ..., e_n^m)$ and $S^{\ell^*} = (s_1^{\ell^*}, ..., s_n^{\ell^*})$, namely:

$$\mathcal{L}_{intermed} = \text{smoothL1}(E^m,\ S^{\ell^*}), \tag{3}$$

where $\text{smoothL1}(x, y)$ equals $\frac{1}{2}(x - y)^2$ if $|x - y| < 1$, and $|x - y| - \frac{1}{2}$ otherwise [12].

### 3.2.1 Loss Functions

As shown in Figure 3 (a), the loss function of DREAM comprises three components. First, we encourage the output features of the draft model to closely match those of the target model by minimizing the difference between the final-layer features $E^M = (e_1^M, \ldots, e_n^M)$ from the draft model and $S^L = (s_1^L, \ldots, s_n^L)$ from the target model, using a smooth L1 loss: $\mathcal{L}_{feat} = \text{smoothL1}(E^L, S^L)$. This will improve the acceptance rate of the draft tokens. Second, following the adaptive feature selection strategy described in Section 3.2, we minimize the difference between an early-layer feature $E^m$ and the selected intermediate target feature $S^{\ell^*}$. For DREAM, we set $m = 1$, yielding the loss: $\mathcal{L}_{intermed} = \text{smoothL1}(E^m, S^{\ell^*})$. Finally, to ensure the token outputs from the draft model $M_{da}$ align with those from the target model $M_{ta}$, we apply a KL divergence loss between their softmax outputs: $\mathcal{L}_{KL} = \text{KL}(\text{softmax}(D), \text{softmax}(T))$, where $D = (d_1, \ldots, d_n)$ and $T = (t_1, \ldots, t_n)$ are the predicted token logits from the draft and target models, respectively. Finally, the overall loss function can be described as:

$$\mathcal{L}_{final} = \lambda_{feat}\,\mathcal{L}_{feat} + \lambda_{intermed}\,\mathcal{L}_{intermed} + \lambda_{KL}\,\mathcal{L}_{KL}, \tag{4}$$

where $\lambda_{feat}$, $\lambda_{intermed}$ and $\lambda_{KL}$ denotes the relative importance between the loss functions.

### 3.3 Visual Token Compression

Given the high computational cost of visual tokens, as discussed in Section 2.2, DREAM adopts subsampling strategy to reduce this overhead. A naive solution would be to uniformly subsample the visual tokens. However, this approach risks discarding crucial visual information necessary for accurate draft model predictions, while retaining redundant regions that could be aggressively compressed or removed. To address this, DREAM incorporates a simple yet effective token selection mechanism, as illustrated in the visual token selection block of Figure 3 (b). Specifically, it computes importance scores for the final-layer features $(s_1^L, \ldots, s_{q+v}^L)$ by summing each token's attention weights across all other tokens, as derived from the attention matrix. These scores reflect the relative importance of each token in the last layer with respect to the final model accuracy.

Among the attention scores, we isolate those corresponding to the visual input and sort this subset by magnitude. We then record the indices of the top scores based on the subsampling ratio $r$. These indices are used to subsample the visual tokens $(s_{q+1}^1, \ldots, s_{q+v}^1)$ with the top high scores, where they are considered critical to the final output. This strategy significantly reduces the number of tokens while retaining the most important visual information. Although visual token subsampling may cause a slight decrease in the draft model's accuracy, our evaluation in Section 4 shows that it can effectively reduce speculative decoding latency.

## 4 Empirical Results

We conduct experiments on five VLMs representing a range of parameter scales, including LLaVA-v1.6-Vicuna (7B, 13B) [30], Pixtral (12B) [1], SmolVLM (2B) [38], and Gemma3 (12B) [53]. DREAM is evaluated across eight diverse benchmarks: MMT-Bench [58], SEED-Bench-2 [21], ScienceQA [37], OCRBench [35], ChartQA [39], and MathVista [36]. All evaluations are performed under two softmax temperature settings: Temp = 0 and Temp = 1. We report two key metrics: (1) **Speedup ratio** over standard autoregressive generation, defined as $t_{\text{AR}}/t_{\text{method}}$, where $t_{\text{AR}}$ is

the average wall-clock time per token for standard decoding, and $t_{\mathrm{method}}$ is the corresponding time for each evaluated method. A larger speedup directly corresponds to lower end-to-end latency in real-world use. (2) **Average token acceptance length** $\tau$, representing the number of consecutive draft tokens accepted by the verification model. A larger $\tau$ implies fewer verification steps and higher effective decoding throughput. We adapt six recent SD baselines, originally developed for text-only LLMs, for use with VLMs, including SPD [10], Kangaroo [28], Medusa [4], Hydra [2], and EAGLE 1 and 2 [25, 24].

We freeze the target VLMs and train only the draft model using the LLaVA `mix665k` dataset, with 55,000 training samples and 1,000 training samples from each corresponding evaluation benchmark dataset. For benchmarks without predefined train–test splits (MMT-Bench, SEED-Bench-2, Math-Vista, and OCRBench), we applied stratified random sampling using `train_test_split` (with `random_state=42` for reproducibility) to select 1,000 training samples and up to 3,000 test samples per dataset. For ChartQA and ScienceQA, which already provide official splits, we randomly sampled 1,000 training examples from their respective training sets. All selected training samples were then processed through the target model to generate ground-truth responses in conversational form.

Training is performed for 68,000 iterations using AdamW optimizer ($\beta_1 = 0.9$, $\beta_2 = 0.95$), a learning rate of $3 \times 10^{-5}$, and gradient clipping set to 0.5. Each draft model consists of an initial decoder block, an intermediate cross-attention block, and a final decoder block. The weights for the loss terms $\mathcal{L}_{\mathrm{feat}}$, $\mathcal{L}_{\mathrm{intermed}}$, and $\mathcal{L}_{\mathrm{KL}}$ are set to 0.2, 0.2, and 1.0. The parameter sizes are 0.65B for LLaVA-v1.6-Vicuna-7B, 0.98B for LLaVA-v1.6-Vicuna-13B, 0.9B for Pixtral-12B, 0.28B for SmolVLM-2B, and 0.9B for Gemma3-12B. Training is conducted on two NVIDIA A100 80 GB GPUs with batch size set to 4. Our training procedure adopts a two-stage pipeline: (1) offline calibration for feature selection, and (2) draft model training. During the calibration stage, we compute the attention entropy for each sample once to identify the optimal intermediate layer $\ell^*$, and cache these selections for subsequent training.

During the DREAM evaluation, we use speculative sampling with a batch size of 1, following prior work. The tree structure is configured with $k = 4$ child nodes, a depth of 6, and a maximum draft length of 32 tokens. Note that the average accepted token length ($\tau$) reported in Table 1 may exceed a depth of 6, as the target model generates an additional token once all drafts are verified, resulting in up to 7 tokens per round [19]. 75% of the visual tokens are retained. These values were chosen based on preliminary experiments to balance aggressive speculation against verification overhead. A full sensitivity analysis and ablation study on these hyperparameters can be found in supplementary materials. All models are evaluated on a single NVIDIA A100 80GB GPU.

All models are based on their official Hugging Face implementations. Baseline methods, including Kangaroo [28], Medusa [4], EAGLE [25] and EAGLE-2 [24], are executed using their publicly released code and default configurations, with minimal modifications to support VLM inputs. During inference, all models are run with KV caching enabled to ensure efficient autoregressive decoding. To ensure a fair comparison, all methods are trained and evaluated using the same dataset, number of training epochs, learning rate, batch size, and hardware environment. We fix random seeds across all runs to reduce performance variance and report the average over three runs.

## 4.1 Evaluation Results

Table 1 presents the speedup ratios and average acceptance lengths $\tau$ for DREAM compared to baseline methods. Across all tasks and target models, DREAM consistently achieves the highest speedup and longest acceptance lengths. Notably, DREAM delivers a $1.5\times$ to $3.6\times$ speedup over standard autoregressive decoding with the target model, and achieves a $20\%$ to $40\%$ improvement over EAGLE-2. Particularly, we observe that larger models gain greater benefits from DREAM. At $T = 0$, DREAM achieves an average speedup of $3.06\times$ on LLaVA-v1.6-Vicuna-7B and $2.65\times$ on Pixtral-12B, whereas the speedups are lower for smaller models, with $2.27\times$ on SmolVLM-2B and $2.23\times$ on LLaVA-v1.6-Vicuna-7B. This is because larger models experience more severe decoding bottlenecks, allowing the draft model to more effectively substitute the costly decoding process of them. Furthermore, models like LLaVA and Pixtral embed visual features directly into the language decoder, offering clearer multimodal cues. This allows DREAM to achieve higher acceptance lengths, for example, $\tau = 5.51$ on LLaVA-v1.6-Vicuna-7B. In contrast, models such as Gemma3-12B, which handle cross-modal information through more complex processing pathways, reach lower acceptance rates, with $\tau = 2.36$.

Table 1: Evaluation of SD methods through speedup ratio (S) and average accepted token length ($\tau$).

| Models | Methods | MMT | | SEED | | ScienceQA | | OCRBench | | ChartQA | | MathVista | | Average | |
|---|---|---|---|---|---|---|---|---|---|---|---|---|---|---|---|
| | | S | $\tau$ | S | $\tau$ | S | $\tau$ | S | $\tau$ | S | $\tau$ | S | $\tau$ | S | $\tau$ |
| | | | | | | | | Temperature = 0 | | | | | | | |
| LLaVA-v1.6 Vicuna-7B | SPD [10] | 1.10 | 1.88 | 0.81 | 1.17 | 1.08 | 1.87 | 0.89 | 1.25 | 0.91 | 1.24 | 1.06 | 1.76 | 0.97 | 1.53 |
| | Kangaroo [28] | 1.32 | 2.11 | 1.33 | 2.12 | 1.31 | 2.09 | 1.17 | 1.89 | 1.18 | 1.98 | 1.15 | 1.86 | 1.24 | 2.01 |
| | Medusa [4] | 1.58 | 2.88 | 1.59 | 3.01 | 1.44 | 2.77 | 1.22 | 2.33 | 1.25 | 2.41 | 1.22 | 2.34 | 1.38 | 2.62 |
| | Hydra [2] | 1.78 | 3.86 | 1.72 | 3.88 | 1.68 | 3.79 | 1.41 | 3.21 | 1.35 | 3.11 | 1.42 | 3.25 | 1.56 | 3.52 |
| | EAGLE [25] | 2.10 | 5.04 | 2.09 | 5.01 | 1.98 | 4.88 | 1.72 | 4.13 | 1.56 | 3.98 | 1.78 | 4.25 | 1.87 | 4.55 |
| | EAGLE-2 [24] | 2.31 | 5.48 | 2.31 | 5.61 | 2.15 | 5.22 | 1.92 | 4.88 | 1.77 | 4.22 | 1.87 | 4.67 | 2.05 | 5.01 |
| | **DREAM** | **2.52** | **6.40** | **2.48** | **6.20** | **2.33** | **5.82** | **2.05** | **4.88** | **1.89** | **4.44** | **2.11** | **5.32** | **2.23** | **5.51** |
| LLaVA-v1.6 Vicuna-13B | SPD | 1.07 | 1.78 | 1.06 | 1.79 | 1.09 | 1.88 | 0.86 | 1.12 | 0.89 | 1.25 | 0.87 | 1.22 | 1.00 | 1.58 |
| | Kangaroo | 1.43 | 1.77 | 1.51 | 1.87 | 1.22 | 1.55 | 1.21 | 1.54 | 1.27 | 1.61 | 1.53 | 2.01 | 1.36 | 1.72 |
| | Medusa | 1.99 | 2.67 | 1.96 | 2.76 | 1.93 | 2.77 | 1.40 | 2.92 | 1.51 | 2.82 | 1.51 | 2.62 | 1.72 | 2.76 |
| | Hydra | 2.12 | 2.87 | 2.08 | 2.99 | 2.21 | 3.12 | 1.49 | 3.07 | 1.65 | 3.03 | 1.66 | 2.87 | 1.87 | 2.99 |
| | EAGLE | 2.45 | 3.56 | 2.19 | 3.24 | 2.63 | 3.98 | 1.65 | 3.31 | 1.85 | 3.27 | 1.8 | 3.09 | 2.10 | 3.41 |
| | EAGLE-2 | 2.89 | 4.05 | 3.18 | 4.33 | 3.09 | 4.97 | 2.20 | 4.12 | 2.41 | 4.15 | 2.39 | 3.76 | 2.69 | 4.23 |
| | **DREAM** | **3.68** | **5.58** | **3.51** | **5.34** | **3.36** | **5.29** | **2.69** | **4.64** | **2.59** | **4.20** | **2.53** | **4.18** | **3.06** | **4.87** |
| Pixtral-12B | SPD | 1.08 | 1.51 | 1.03 | 1.47 | 1.05 | 1.49 | 1.05 | 1.49 | 1.04 | 1.43 | 1.04 | 1.46 | 1.05 | 1.47 |
| | Kangaroo | 1.26 | 1.54 | 1.09 | 1.39 | 1.14 | 1.51 | 1.16 | 1.52 | 1.12 | 1.47 | 1.13 | 1.49 | 1.15 | 1.49 |
| | Medusa | 1.37 | 1.81 | 1.37 | 1.81 | 1.35 | 1.87 | 1.24 | 1.69 | 1.22 | 1.68 | 1.16 | 1.47 | 1.28 | 1.72 |
| | Hydra | 1.58 | 2.24 | 1.47 | 2.04 | 1.53 | 2.06 | 1.38 | 1.81 | 1.34 | 1.79 | 1.36 | 1.78 | 1.44 | 1.95 |
| | EAGLE | 2.38 | 3.47 | 1.97 | 2.53 | 2.31 | 3.64 | 1.69 | 2.73 | 1.78 | 2.84 | 1.64 | 2.47 | 1.96 | 2.95 |
| | EAGLE-2 | 2.81 | 3.95 | 2.31 | 3.07 | 2.64 | 4.03 | 2.12 | 3.25 | 2.14 | 3.17 | 1.81 | 2.73 | 2.31 | 3.37 |
| | **DREAM** | **2.93** | **4.52** | **2.61** | **3.67** | **2.98** | **4.33** | **2.38** | **3.55** | **2.35** | **3.49** | **2.36** | **3.42** | **2.65** | **3.78** |
| SmolVLM-2B | SPD | 1.02 | 1.33 | 1.04 | 1.41 | 1.06 | 1.43 | 1.06 | 1.42 | 1.07 | 1.46 | 1.02 | 1.34 | 1.04 | 1.40 |
| | Kangaroo | 1.28 | 1.48 | 1.08 | 1.18 | 1.03 | 1.17 | 1.06 | 1.22 | 1.04 | 1.14 | 1.08 | 1.23 | 1.10 | 1.24 |
| | Medusa | 2.12 | 2.71 | 1.51 | 2.00 | 1.72 | 2.22 | 1.20 | 1.61 | 1.15 | 1.55 | 1.35 | 1.75 | 1.51 | 1.97 |
| | Hydra | 2.33 | 3.07 | 1.62 | 2.08 | 1.98 | 2.66 | 1.32 | 1.74 | 1.22 | 1.58 | 1.51 | 1.98 | 1.66 | 2.19 |
| | EAGLE | 2.57 | 3.42 | 1.85 | 2.56 | 2.16 | 2.76 | 1.42 | 1.88 | 1.34 | 1.77 | 1.65 | 2.22 | 1.83 | 2.44 |
| | EAGLE-2 | 2.96 | 3.89 | 2.12 | 2.93 | 2.39 | 3.21 | 1.65 | 2.11 | 1.51 | 2.13 | 1.81 | 2.63 | 2.07 | 2.82 |
| | **DREAM** | **3.05** | **3.97** | **2.24** | **3.18** | **2.85** | **3.62** | **1.85** | **2.56** | **1.62** | **2.33** | **2.01** | **2.88** | **2.27** | **3.09** |
| Gemma3-12B | Kangaroo | 1.37 | 1.66 | 1.47 | 1.79 | 1.52 | 1.57 | 3.17 | 2.28 | 2.28 | 1.85 | 1.18 | 1.64 | 1.83 | 1.80 |
| | EAGLE | 1.73 | 1.98 | 1.69 | 2.52 | 1.72 | 1.97 | 4.26 | 2.42 | 3.40 | 1.99 | 1.42 | 1.89 | 2.37 | 2.13 |
| | EAGLE-2 | 2.92 | 1.99 | 1.74 | 2.79 | 1.92 | 1.98 | 4.68 | 2.57 | 3.48 | 2.23 | 1.52 | 1.91 | 2.71 | 2.25 |
| | **DREAM** | **2.99** | **2.13** | **3.53** | **2.84** | **2.60** | **2.05** | **4.81** | **2.58** | **3.68** | **2.56** | **1.98** | **1.99** | **3.27** | **2.36** |
| | | | | | | | | Temperature = 1 | | | | | | | |
| LLaVA-v1.6 Vicuna-7B | SPD | 0.83 | 1.19 | 0.81 | 1.15 | 0.85 | 1.18 | 0.75 | 1.06 | 0.72 | 1.08 | 0.92 | 1.48 | 0.81 | 1.19 |
| | Kangaroo | 1.20 | 1.97 | 1.26 | 2.03 | 1.23 | 2.01 | 1.09 | 1.80 | 1.11 | 1.89 | 1.07 | 1.77 | 1.16 | 1.91 |
| | EAGLE-2 | 2.19 | 5.37 | 2.20 | 5.48 | 2.04 | 5.12 | 1.82 | 4.77 | 1.65 | 4.13 | 1.76 | 4.56 | 1.95 | 4.91 |
| | **DREAM** | **2.39** | **6.29** | **2.35** | **6.07** | **2.25** | **5.68** | **1.99** | **4.88** | **1.84** | **4.41** | **2.02** | **5.23** | **2.14** | **5.43** |
| LLAVA-v1.6 Vicuna-13B | SPD | 0.88 | 1.22 | 0.84 | 1.25 | 0.84 | 1.32 | 0.79 | 1.18 | 0.81 | 1.14 | 0.88 | 1.24 | 0.84 | 1.22 |
| | Kangaroo | 1.23 | 1.57 | 1.17 | 1.53 | 1.07 | 1.44 | 1.01 | 1.24 | 1.07 | 1.34 | 1.21 | 1.67 | 1.13 | 1.46 |
| | EAGLE-2 | 2.35 | 3.75 | 3.02 | 4.30 | 3.03 | 4.67 | 2.03 | 3.87 | 2.18 | 3.83 | 2.18 | 3.41 | 2.46 | 3.97 |
| | **DREAM** | **3.34** | **5.38** | **3.32** | **5.06** | **3.20** | **5.98** | **2.22** | **3.89** | **2.43** | **4.04** | **2.29** | **4.03** | **2.80** | **4.73** |
| Pixtral-12B | SPD | 0.81 | 1.15 | 0.79 | 1.11 | 0.80 | 1.12 | 0.80 | 1.13 | 0.75 | 1.07 | 0.77 | 1.09 | 0.79 | 1.11 |
| | Kangaroo | 1.18 | 1.41 | 1.08 | 1.35 | 1.03 | 1.36 | 1.19 | 1.48 | 1.14 | 1.45 | 1.09 | 1.41 | 1.12 | 1.41 |
| | EAGLE-2 | 2.76 | 3.81 | 2.24 | 3.01 | 2.76 | 3.87 | 2.23 | 3.24 | 2.03 | 3.09 | 1.79 | 2.69 | 2.30 | 3.28 |
| | **DREAM** | **2.90** | **4.02** | **2.47** | **3.57** | **2.93** | **3.94** | **2.29** | **3.46** | **2.21** | **3.21** | **2.16** | **3.27** | **2.49** | **3.58** |
| SmolVLM-2B | SPD | 1.07 | 1.47 | 1.01 | 1.33 | 1.07 | 1.46 | 0.97 | 1.26 | 1.06 | 1.44 | 0.85 | 1.20 | 1.00 | 1.36 |
| | Kangaroo | 1.37 | 1.59 | 1.12 | 1.24 | 1.22 | 1.41 | 1.12 | 1.29 | 1.18 | 1.36 | 1.28 | 1.42 | 1.22 | 1.39 |
| | EAGLE-2 | 2.62 | 3.60 | 1.92 | 2.67 | 2.24 | 3.11 | 1.41 | 1.77 | 1.77 | 2.18 | 1.77 | 2.49 | 1.93 | 2.64 |
| | **DREAM** | **2.88** | **3.66** | **2.25** | **3.33** | **2.91** | **3.74** | **1.54** | **2.12** | **1.77** | **2.51** | **1.97** | **2.70** | **2.22** | **3.01** |
| Gemma3-12B | Kangaroo | 1.83 | 1.66 | 1.23 | 2.61 | 1.56 | 2.29 | 3.34 | 2.27 | 2.23 | 1.86 | 1.16 | 1.65 | 1.89 | 2.06 |
| | EAGLE | 2.23 | 1.96 | 1.60 | 2.52 | 2.16 | 1.97 | 3.74 | 2.65 | 3.30 | 2.03 | 1.59 | 1.86 | 2.44 | 2.16 |
| | EAGLE-2 | 2.73 | 1.94 | 2.13 | 2.79 | 2.21 | 2.07 | 4.67 | 2.47 | 3.35 | 2.23 | 1.65 | 1.89 | 2.79 | 2.23 |
| | **DREAM** | **2.88** | **2.07** | **3.49** | **2.84** | **2.39** | **2.12** | **4.79** | **2.56** | **3.61** | **2.43** | **1.96** | **1.91** | **3.19** | **2.32** |

Second, the task-level analysis reveals that long-form QA tasks such as MMT-Bench and ScienceQA benefit most from speculative decoding. These tasks require generating structured answers grounded in high-level visual semantics, which DREAM captures efficiently. For example, on ScienceQA with LLaVA-v1.6-Vicuna-7B, DREAM achieves a speedup of 3.36× with $\tau = 5.32$, outperforming Kangaroo (1.22×) and EAGLE-2 (3.09×). In contrast, OCR-style datasets such as MathVista pose greater challenges due to their reliance on fine-grained character-level recognition, which current draft models struggle to replicate. As a result, all methods show lower $\tau$ and speedup on this benchmark.

Finally, temperature settings critically impact performance. At low temperature ($T = 0$), deterministic decoding leads to higher token alignment and longer acceptance spans. For instance, DREAM achieves $\tau = 4.52$ on MMT-Bench with Pixtral-12B. When sampling is enabled ($T = 1$), perfor-

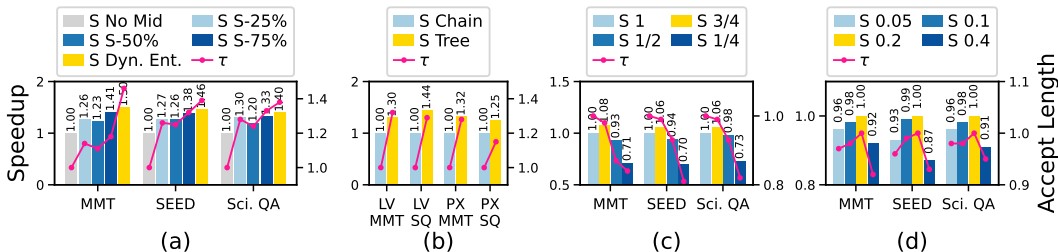

Figure 4: Normalized speedup S and normalized accepted token length $\tau$ across (a) intermediate feature selection strategies. (b) chain-based and tree-based decoding. (c) visual token compression ratios, where 1 and 3/4 denote 100% and 75% of the visual tokens are retained, respectively. (d) loss weight settings, where the number is the value for $\lambda_{feat}$ and $\lambda_{intermed}$. $\lambda_{KL}$ is fixed to 1.

mance moderately degrades due to increased token variance, yet DREAM's tree-based architecture maintains robust performance. On MMT-Bench with Pixtral-12B, DREAM still delivers a speedup of 2.90× and $\tau = 4.02$, outperforming all baselines under the same temperature.

## 4.2 Ablation Study

**Impact of Draft Model Architecture** In this section, we evaluate the impact of draft model architecture by testing DREAM on LLaVA-v1.6-Vicuna-7B across multiple datasets, with VTC disabled for all. We design six variants of the DREAM draft model to assess architectural contributions. In the first three baseline configurations, we remove one component at a time: the initial decoder block, denoted as *w/o Initial*, the cross-attention block (*w/o CA*), or the final decoder block (*w/o Final*). Additionally, we include a variant with an extra cross-attention block to evaluate the effect of deeper feature fusion, denoted as *2 CA*. We further evaluate EAGLE-2 by varying its draft model depth, using one, three, and four decoder blocks, denoted as *E2-1B* (default setting in EAGLE-2), *E2-3B*, and *E2-4B*, respectively. Table 2 summarizes the results. Notably, the removal of the cross-attention block causes the most substantial performance drop, showing its essential role in the draft model architecture. Adding an extra cross-attention block increases the accepted sequence length, indicating improved draft quality. However, the larger draft model also incurs additional latency, which offsets the overall speedup gains. Similarly, increasing the number of blocks in the EAGLE-2 draft model improves the average acceptance length, but speedup diminishes once the depth reaches four blocks. Overall, DREAM outperforms EAGLE-2 in both speedup and acceptance length, highlighting the importance of the cross-attention block in DREAM.

Table 2: Comparison over different model architectures. Numbers are normalized to DREAM results.

| Baseline | MMT-Bench | | SEED-Bench | | ScienceQA | |
| --- | --- | --- | --- | --- | --- | --- |
| | Speedup | $\tau$ | Speedup | $\tau$ | Speedup | $\tau$ |
| DREAM | 1 | 1 | 1 | 1 | 1 | 1 |
| w/o Initial | 0.60 | 0.59 | 0.60 | 0.58 | 0.64 | 0.61 |
| w/o CA | 0.48 | 0.44 | 0.46 | 0.41 | 0.41 | 0.32 |
| w/o Final | 0.73 | 0.63 | 0.71 | 0.65 | 0.67 | 0.71 |
| 2 CA | 0.99 | 1.09 | 1.02 | 1.13 | 0.79 | 1.10 |
| E2-1B | 0.82 | 0.71 | 0.81 | 0.77 | 0.80 | 0.75 |
| E2-3B | 0.88 | 0.86 | 0.89 | 0.88 | 0.91 | 0.88 |
| E2-4B | 0.63 | 0.92 | 0.63 | 0.92 | 0.68 | 0.93 |

**Impact of Intermediate Feature Selection** In this section, we evaluate DREAM's intermediate feature selection strategy for draft model training, referred to as *Dynamic Entropy* (Dyn. Ent.), against four baseline methods on the LLaVA-v1.6-Vicuna-7B model across the MMT-Bench, SEED-Bench, and ScienceQA datasets. The first baseline, *No Mid Tuning* (No Mid), trains the draft model without using any intermediate features. The second baseline, *Static-25%* (S-25%), uses intermediate features from the 25% depth of the target model (the 10th layer in LLaVA-v1.6-Vicuna-7B) for training. Similarly, *Static-50%* (S-50%) and *Static-75%* (S-75%) use features from the 50% and 75% depths, corresponding to the 20th and 30th layers, respectively. Figure 4 (a) presents the results. The draft model trained without intermediate features achieves limited speedup, as fewer tokens are accepted. *Static-25%* and *Static-50%* yield comparable speedup, while *Static-75%* performs best among the static approaches, suggesting that deeper intermediate layers provide more informative guidance. *Dynamic Entropy* applied by DREAM outperforms all baselines with the highest speedup.

**Impact of Tree Structure** As shown by prior work, tree-structured drafts improve the accepted sequence length compared to chain-structured drafts [25, 24, 6]. Figure 4 (b) shows the performance of DREAM with both chain- and tree-structured drafts on the LLaVA-v1.6-Vicuna-7B (LV) and Pixtral-12B (PX) models across the MMT-Bench (MMT) and ScienceQA (SQ) datasets. Tree decoding provides an average of $1.32\times$ speedup on top of our proposed methods described in Section 3. It is important to note that all the baseline algorithms we compared except Kangaroo utilize tree-structured decoding. Even without the help of tree decoding, compared with Kangaroo, DREAM achieves an average of 56% and 75% speedup on LLaVA and Pixtral, respectively.

**Impact of Visual Token Compression Rate** To improve decoding efficiency, DREAM subsamples and retains only a fraction of visual tokens from the ViT encoder, as detailed in Section 3.3. In Figure 4 (c), we evaluate the effect of different visual token compression ratios, where a fraction of 1 indicates no compression, and 3/4 means 75% of the tokens are kept. Experiments on the LLaVA-v1.6-Vicuna-7B model across MMT-Bench, SEED-Bench, and ScienceQA show that retaining 3/4 of the tokens yields 7% speedup with only a minor reduction in acceptance length. However, when only 1/2 and 1/4 of the tokens are retained, the reduced visual input leads to noticeable information loss, lowering acceptance length and diminishing the speedup.

Specifically, we examine how the visual token compression (VTC) rate affects both inference speed and draft token accuracy in DREAM. We test four compression levels (100%, 75%, 50%, and 25%) and evaluate their effects using two key metrics: the speedup ratio ($S$) and the average accepted token length ($\tau$) across three benchmarks: MMT-Bench, SEED-Bench, and ScienceQA.

As presented in Table 3, the results show that a 75% VTC rate

Table 3: Effect of Visual Token Compression Rate on Speedup and Token Acceptance

| Models | VTC Rate | MMT-Bench | | SEED-Bench | | ScienceQA | |
|---|---|---|---|---|---|---|---|
| | | $S$ | $\tau$ | $S$ | $\tau$ | $S$ | $\tau$ |
| LLaVA-v1.6 Vicuna-13B | 100% | 3.53 | 5.67 | 3.32 | 5.40 | 3.18 | 5.33 |
| | 75% | 3.68 | 5.58 | 3.51 | 5.34 | 3.36 | 5.29 |
| | 50% | 3.19 | 4.96 | 3.13 | 5.04 | 3.11 | 4.98 |
| | 25% | 2.44 | 4.78 | 2.31 | 4.38 | 2.31 | 4.36 |
| Pixtral-12B | 100% | 2.88 | 4.48 | 2.55 | 3.87 | 2.91 | 4.52 |
| | 75% | 2.93 | 4.52 | 2.61 | 3.67 | 2.98 | 4.03 |
| | 50% | 1.51 | 2.68 | 1.58 | 2.83 | 1.53 | 2.77 |
| | 25% | 1.24 | 2.18 | 1.51 | 2.35 | 1.33 | 2.64 |

generally offers the best balance between speed and accuracy. For LLaVA-13B and Pixtral-12B, this setting achieves the highest or second-highest speedups ($3.68\times$ and $2.93\times$ on MMT-Bench, respectively) while keeping $\tau$ within about 2% of the full-token baseline. In comparison, reducing the visual tokens to 50% or 25% yields only minor speed improvements but causes $\tau$ to drop by 10–25%, indicating that excessive visual compression leads to more draft token rejections due to reduced contextual information.

**Impact of Lambda Settings** The weights $\lambda_{feat}$, $\lambda_{intermed}$, and $\lambda_{KL}$ balance different supervisory signals. Since $\mathcal{L}_{feat}$ and $\mathcal{L}_{intermed}$ are smooth L1 losses of similar scale and play comparable roles in guiding the model by promoting feature alignment, we set $\lambda_{feat} = \lambda_{intermed}$ to simplify tuning. The KL loss is typically smaller in magnitude and is fixed at $\lambda_{KL} = 1$ to maintain consistent influence. As shown in Figure 4 (d), on LLaVA-v1.6-Vicuna-7B, increasing $\lambda_{feat}$ from 0.05 to 0.2 improves speedup and average accepted token length. However, at 0.4, both metrics drop, indicating that excessive feature supervision may harm generalization.

## 5  Conclusion and Limitation

We present DREAM, a speculative decoding framework optimized for VLMs. By integrating visual token compression, cross-attention feature fusion, and adaptive intermediate distillation, DREAM achieves up to $3.6\times$ speedup over standard decoding while maintaining high accuracy, offering a scalable and efficient solution for fast multimodal inference.

Although DREAM shows strong performance, its evaluation is limited to NVIDIA GPUs. As future work, it is important to assess its effectiveness across a broader range of hardware platforms, datasets, and VLMs. Moreover, although DREAM improves the efficiency of VLM, enabling faster and more accessible deployment in real-world applications such as assistive technologies and interactive agents. However, care should be taken to prevent misuse in generating harmful multimodal content.

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
