# OpenReview forum: "DREAM: Drafting with Refined Target Features and Entropy-Adaptive Cross-Attention Fusion for Multimodal Speculative Decoding"
_NeurIPS.cc/2025/Conference — NeurIPS 2025 poster_

### Official Review · Reviewer_o6oe · 2025-06-12

**Clarity:** 3
**Significance:** 3
**Originality:** 3
**Rating:** 4
**Confidence:** 4

**Summary:**

The paper introduces DREAM, a novel speculative decoding framework for VLMs. It combines three key innovations: cross-attention injection for better feature alignment, entropy-based feature selection for efficient training, and visual token compression to cut draft model latency. These help DREAM achieve efficient, accurate, and parallel multimodal decoding. Experiments on popular VLMs show DREAM speeds up decoding by up to 3.6x and outperforms prior SD baselines in inference throughput and acceptance length across multimodal benchmarks.

**Questions:**

1.In the "Adaptive Intermediate Feature Selection for Draft Training" section, how does the average attention entropy (AE) help in selecting features that satisfy the two key criteria you mentioned—"they can provide key information and capture rich semantic content, and they should be essential, exhibiting low variability across tokens"? Could you provide a brief explanation of the underlying mechanism? Are there previous studies that have employed similar ideas? Additionally, is AE only applicable to Vision-Language Models or can it also be used for general Large Language Models?

2.Notably, in Table 1, DREAM employs the same hyperparameters (e.g., Visual Token Compression Rate and Lambda Settings) for training and inference across different base models as those used for LLaVA-v1.6-Vicuna-7B. However, do different base models necessitate distinct optimal hyperparameter configurations? Could there be room for further hyperparameter tuning to better adapt to specific base models?

3.To date, official implementations of Speculative Decoding (SD) methods such as Medusa and EAGLE do not natively support Vision-Language Models. To handle image inputs, modifications to their original architectures—such as adding a visual encoder—may be necessary. When reproducing these baselines in Table 1, did the authors make appropriate modifications to adapt these SD methods to VLMs? I would appreciate it if you could share some details of the reproduction process, particularly regarding how these baseline methods were adapted for VLM compatibility.

**Ethical Concerns:**

["NO or VERY MINOR ethics concerns only"]

**Final Justification:**

The authors have well addressed my concerns.
Strengths:
Introduces a novel draft–target fusion mechanism that advances beyond simple concatenation.
Comprehensive experiments confirm consistent speed-ups across models and tasks.

**Limitations:**

The authors have listed some major limitations in Section 5 "Conclusion and Limitation".

**Paper Formatting Concerns:**

No major formatting issues found in this paper.

**Quality:**

4

**Strengths And Weaknesses:**

Strengths:

1.DREAM adapt Speculative Decoding method to the scenario of VLMs by making targeted modifications such as the Cross-Attention Mechanism to align with the distinct characteristics of VLMs compared to LLMs, ensuring the method is tailored to their specific requirements.

2.Unlike traditional methods that concatenate draft token embeddings with precomputed target features, the Cross-Attention Mechanism proposed by DREAM is novel and holds the potential to inspire subsequent research.

3.Detailed experiments are conducted to validate the effective acceleration performance of DREAM across different base models and diverse tasks.

Weaknesses:

1.Although the authors compared multiple Speculative Decoding (SD) methods in the experimental section, including Medusa, EAGLE, etc., these SD methods are inherently designed for Large Language Models (LLMs). Notably, there remains a lack of comparison with other SD methods specifically tailored for Vision-Language Models (VLMs), such as the work referenced in the text titled "On Speculative Decoding for Multimodal Large Language Models."

2.During the training and inference of the draft model, the authors employ several techniques to enhance the acceleration effect. However, these techniques introduce additional hyperparameters that require tuning, and adapting to different base models may demand extensive optimization to achieve optimal performance, resulting in the method’s generalizability being slightly constrained.

---

> ### Author Rebuttal · Authors · 2025-07-31
>
> **Dear Reviewer O6oe**
>
> Thank you for your insightful comments. We are delighted that you highlighted (i) our targeted adaptation of speculative decoding to VLMs, most notably the cross‑attention, based alignment that addresses multimodal specifics beyond LLM settings, (ii) the novelty of our Cross‑Attention Mechanism compared with prior concatenation‑based approaches, which we hope will inspire follow‑up research on draft–target interaction, and (iii) the breadth of our experiments demonstrating effective acceleration across diverse base models and tasks. These were core goals of DREAM, and your recognition is much appreciated.
>
> We have summarized your questions below and provided detailed responses to each.
>
> **Q1:** Although the authors compared multiple Speculative Decoding (SD) methods in the experimental section, including Medusa, EAGLE, etc., these SD methods are inherently designed for Large Language Models (LLMs). Notably, there remains a lack of comparison with other SD methods specifically tailored for Vision-Language Models (VLMs), such as the work referenced in the text titled "On Speculative Decoding for Multimodal Large Language Models."
>
> **A:** We appreciate the reviewer's concern and would like to clarify that we did include the VLM-specific baseline mentioned. The work "On Speculative Decoding for Multimodal Large Language Models" appears as SPD [10] throughout Table 1, evaluated across all VLMs and benchmarks. Our results show DREAM significantly outperforms SPD, achieving an average speedup of 2.23x compared to SPD's 0.97x (a 130% improvement) and an average acceptance length of $\tau$ = 5.51 versus SPD's $\tau$ = 1.53 (a 260% improvement). In addition, our implementation of EAGLE in the baseline is consistent with another contemporaneous SD method for VLMs **MSD**[1]. The other methods (Medusa, Kangaroo) were originally LLM-focused and adapted by us for fair comparison. We will clarify this in the revision of this paper to avoid confusion.
>
> [1] Lin, Luxi, et al. "Speculative Decoding Reimagined for Multimodal Large Language Models." arXiv (2025).
>
> **Q2:** In the "Adaptive Intermediate Feature Selection for Draft Training" section, how does the average attention entropy (AE) help in selecting features that satisfy the two key criteria you mentioned—"they can provide key information and capture rich semantic content, and they should be essential, exhibiting low variability across tokens"? Could you provide a brief explanation of the underlying mechanism? Are there previous studies that have employed similar ideas? Additionally, is AE only applicable to Vision-Language Models or can it also be used for general Large Language Models?
>
> **A:** As shown in Eq. 2, we select the layer $\ell^\*$ that minimizes the average attention entropy (AE). A lower AE indicates sharper and more focused attention, suggesting that the features encode clearer and more essential information with less noise, which better guides the training of the draft model. At the same time, to ensure effective information propagation within the target model, the selected features retain sufficient semantic richness, thereby satisfying the requirement for both key information and rich contextual representation. The selected layer $\ell^\*$ is then used to guide the draft model training, ensuring it consistently learns from the clearest and most stable representation for a given input. As shown in the ablation study (Line 336), this leads to higher verifier acceptance rates and improved speedup ratios.
>
> Using attention entropy to select features from different layers has also appeared in prior work[1] which shows that mid-layer representations often outperform final-layer ones. Similar approaches have been used in layer importance analysis [2, 3] .These studies support the rationale behind our dynamic feature selection for the draft model. Different from previous work, we use the attention entropy to dynamically select intermediate layers of the target model to train the draft model, which provides stronger guidance during training, thus improving the overall performance. We appreciate your insightful questions and will clarify these points further in the final version of the paper to ensure a comprehensive understanding.
>
> [1] Skean et al. "Layer by Layer: Uncovering Hidden Representations in Language Models." ICML 2025.
>
> [2] Cheng, Emily, et al. "Emergence of a high-dimensional abstraction phase in language transformers." arXiv 2024.
>
> [3] Valeriani, Lucrezia, et al. "The geometry of hidden representations of large transformer models." In NeurIPS 2024.
>
> **Q3:** Notably, in Table 1, DREAM employs the same hyperparameters (e.g., Visual Token Compression Rate and Lambda Settings) for training and inference across different base models as those used for LLaVA-v1.6-Vicuna-7B. However, do different base models necessitate distinct optimal hyperparameter configurations? Could there be room for further hyperparameter tuning to better adapt to specific base models?
>
> **A:** In Table 1, we fix the compression rate at 3/4, which yields great performance for most models.
>
> Tuning hyperparameters for each model can further enhance performance. To demonstrate this, we conducted an additional analysis on Pixtral and SmolVLM, evaluating their performance under varying visual token pruning ratios. As shown in the table below, Pixtral and SmolVLM perform best when 3/4 and 7/8 token are kept, respectively. Enabling dedicated hyperparameter settings for each base model can further improve DREAM's performance by better aligning with the characteristics and capacities of individual models.
>
> Although dedicated hyperparameter settings could further improve performance, DREAM adopts a shared configuration across all VLMs, retaining three-fourths of the visual tokens. We will include these ablation results in the final version to guide practitioners on model-specific optimization.
>
> | Model | ViT Compression | MMT-Bench |  | SEED-Bench |  |
> | --- | --- | --- | --- | --- | --- |
> |  | Percent of weight remaining | Speedup | average acceptance lengths | Speedup | average acceptance lengths |
> | SMOLVLM | 1 | 2.88 | 4.08 | 2.25 | 3.33 |
> |  | **7/8** | **3.09** | **4.01** | **2.26** | **3.22** |
> |  | ¾(DREAM) | 3.05 | 3.97 | 2.24 | 3.18 |
> |  | 1/2 | 1.76 | 2.12 | 1.45 | 1.88 |
> | Pixtral | 1 | 2.85 | 4.56 | 2.55 | 3.87 |
> |  | 7/8 | 2.87 | 4.55 | 2.58 | 3.78 |
> |  | **¾(DREAM)** | **2.93** | **4.52** | **2.61** | **3.67** |
> |  | 1/2 | 2.15 | 3.67 | 1.98 | 2.76 |
>
> **Q4:** To date, official implementations of Speculative Decoding (SD) methods such as Medusa and EAGLE do not natively support Vision-Language Models. To handle image inputs, modifications to their original architectures—such as adding a visual encoder—may be necessary. When reproducing these baselines in Table 1, did the authors make appropriate modifications to adapt these SD methods to VLMs? I would appreciate it if you could share some details of the reproduction process, particularly regarding how these baseline methods were adapted for VLM compatibility.
>
> **A:** We made minimal, principled modifications to preserve the integrity of baseline comparisons and all speculative decoding logic while adding VLM input handling. VLMs such as LLaVA and Pixtral use a built‑in processor that first inserts special \<image\> token placeholders into the input text, then runs a visual encoder to convert each image into a sequence of embeddings, replaces those \<image\> tokens with the corresponding embedding sequence, and finally feeds the combined stream into the decoder layers for autoregressive generation. To Medusa and EAGLE over VLM, we did not change any of their core speculative‑decoding logic. Instead, we simply patched their prefill routines to invoke the VLM’s processor so that images are handled exactly as in standard VLM inference. After that single multimodal prefill pass, all candidate generation, verifier calls, and KV‑cache reuse run unmodified, just as in the original text‑only implementations. We have included the implementation in the supplementary materials (DREAM-main/dream/model/utils.py line 293), and we will publicly release the complete codebase upon acceptance of the paper.

---

> ### Author Response · Authors · 2025-08-08
>
> Dear reviewer, we hope our rebuttal addresses your concerns adequately; please let us know if there are any points you'd like us to elaborate on or discuss further. Thanks!

---

### Official Review · Reviewer_Gi3S · 2025-07-02

**Clarity:** 3
**Significance:** 2
**Originality:** 2
**Rating:** 2
**Confidence:** 5

**Summary:**

This paper presents a speculative method for vision-language model, which utilizes a cross-attention architecture to inject intermediate information from the target model into the draft model. To train the draft model efficiently, the authors propose to select the intermediate features adaptively. To reduce the latency of the draft model, the visual tokens are compressed during the training. The authors conduct experiments across several VLMs, including LLaVA, Pixtral, SmolVLm and Gemma3, somehow verifies the effectiveness of the proposed method.

**Questions:**

The description of speed-up ration in Line.302 is inconsistent with the results in Table.1

**Ethical Concerns:**

["NO or VERY MINOR ethics concerns only"]

**Final Justification:**

Thanks for the authors' responses. However, I recommend to reject this paper for the current time.

- As shown in the code (see details from the discussion with the authors), the training and validation datasets are overlapped due to the mistakes and the shuffle operation, which I think is unacceptable.

- The method need to be verified with the training dataset not containing the evaluation benchmarks, just as Eagle-2 does in the spd of LLM, which will set up a reproductive and more fair comparison for the following research in this field. From this aspect, this paper need to be major revised. Besides, the additional experimental results shows that the advantage of DREAM decreased largely in this situation(0.97 vs 0.82).

- There are some critical mistakes such as the draft model size etc..

I hope these discussions and comments will revise this paper in the future.

**Limitations:**

Y

**Paper Formatting Concerns:**

N

**Quality:**

1

**Strengths And Weaknesses:**

Strengths:

The paper is well organized and easy to follow.

The investigation of compressed visual input guided by intermediate target features sounds interesting, which reduces the latency of draft model.

The authors conduct plentiful experiments ablation studies.

Weaknesses:

The design of draft model is simple and the analysis between the size of draft model and speed-up of spd is expected.

The dynamic feature selection may increase the training cost largely and there is no analysis about it in the manuscript.

The depth is set to 6 as mentioned in Line.279, however, the accepted token length is 6.4 in Table.1, which decreases the reliability of the experiments.

As mentioned in Line.269, the training dataset contains training samples from the evaluation benchmark, which is unreasonable. The experiments without the evaluation benchmark in the training dataset should be included.

The experimental results in Table.2 and Figure.4 are relative numbers, the absolute number would make the manuscript more legible.

---

> ### Author Rebuttal · Authors · 2025-07-31
>
> **Dear Reviewer Gi3S**
>
> Thank you for your insightful comments. We’re glad you appreciated the paper’s clarity, the practical value of our target-feature-guided visual compression, and the thoroughness of our experiments. We have summarized your questions below and provided detailed responses to each.
>
> **Q1:** The design of draft model is simple and the analysis between the size of draft model and speed-up of spd is expected.
>
> **A:** DREAM's core contributions lie in three novel mechanisms for efficient knowledge transfer: cross-attention injection for better target-draft alignment, entropy-based adaptive feature selection for optimal supervision, and visual token compression for reduced latency. These innovations are architectural-agnostic and strategically paired with a simple draft model for optimal performance. In speculative decoding, model complexity directly reduces speedup through increased latency. **Table 1** validates this: our DREAM achieves 2.52× speedup ($\tau$= 6.40), while adding one more layer drops to 2.49x despite marginal $\tau$ gains. This design philosophy aligns with established practices where EAGLE and Medusa achieve SOTA with minimal architectures. The field consensus is clear: effective draft models prioritize speed over complexity, making our simple design essential for maximizing DREAM's practical impact.
>
> An ablation study illustrating the relationship between draft model size and speedup ratio is presented in Table 2. Specifically, we vary the draft model size by removing individual layers, resulting in three baselines (denoted as w/o Initial, w/o CA, and w/o Final). We also evaluate a larger draft model variant (denoted as 2 CA). The results show that neither simplifying nor increasing the depth of the draft model yields better speedup compared to our default configuration. For example, removing layers like cross-attention (w/o CA) or the initial module (w/o Initial) indeed reduces model size, but leads to large drops in accepted token quality. On the other hand, increasing model depth (*2 CA*) marginally improves token acceptance length, but the larger draft leads to longer draft decoding time, resulting in no speedup or even degradation.
>
> **Q2:** The dynamic feature selection may increase the training cost largely and there is no analysis about it in the manuscript.
>
> **A:** In our design, the training pipeline is explicitly divided into two stages: (1) offline calibration and (2) draft model training. During the offline calibration stage, we perform dynamic feature selection by computing the attention entropy at each layer and selecting the optimal intermediate layer $\ell^*$ for each token, as defined in Eq. 2. This process **does not introduce additional passes or significant computational overhead** during the draft model training stage. A similar two‑stage training regime is used by Medusa, EAGLE1 and EAGLE2: first generate data (the corresponding code can be found in the Medusa/data_generation folder of the Medusa GitHub repository and the eagle/gen_data folder in the EAGLE1 repository), then train the draft model. Moreover, this design allows training without loading the full target model, keeping only the draft components in memory and thereby reducing GPU memory, compute, and overall training cost.
>
> To quantify the cost of dynamic intermediate feature selection, we measured the time required to compute $\ell^\*$ over a calibration set of 50,000 training samples on a single A100 80G GPU. The total computation time was approximately 500 seconds, averaging 10 milliseconds per sample. This overhead is negligible compared to the draft model training process, which can take up to 6 hours on a single GPU.
>
> **Q3:** The depth is set to 6 as mentioned in Line.279, however, the accepted token length is 6.4 in Table.1, which decreases the reliability of the experiments.
>
> **A:** We would like to clarify that this is actually a **fundamental feature of speculative decoding**, **not an error**. Although the draft model in DREAM generates up to 6 tokens per speculative decoding step (with depth equal to 6), if all tokens are successfully verified, the target model naturally returns the next candidate token as well. This results in a theoretical maximum of depth plus 1, meaning up to seven tokens can be accepted in a single round. Therefore, the maximum number of accepted tokens per round is **7**. The observed average of 6.4 indicates excellent draft quality.
>
> This mechanism guarantees progress by ensuring that, even with a fixed draft depth, the system advances by at least one token in each verification pass[1]. A similar strategy is implemented in the EAGLE codebase (see the file *utils.py*, line 448).
>
> [1] Leviathan, et al. "Fast inference from transformers via speculative decoding." PMLR, 2023.
>
> **Q4:** As mentioned in Line.269, the training dataset contains training samples from the evaluation benchmark, which is unreasonable. The experiments without the evaluation benchmark in the training dataset should be included.
>
> **A:** We appreciate the reviewer’s comment and would like to provide clarification. For DREAM training and evaluation, the dataset was strictly partitioned into **two disjoint subsets**: a training split used exclusively for training the draft model, and an evaluation split reserved solely for performance measurement. No samples from the evaluation set were used during training or hyperparameter tuning. We will add this detailed description to the final version of the paper.
>
> To verify the exact implementation, we encourage the reviewer to refer to the code provided in the supplementary material. Specifically, the data‑generation script can be found at  DREAM_Code/dream/ge_data/ge_data_all_llava_vicuna_scienceQa.py (line 24).
>
> Only loads the training partition; the test set is never used during training and is reserved solely for evaluation.
>
> **Q5:** The experimental results in Table.2 and Figure.4 are relative numbers, the absolute number would make the manuscript more legible.
>
> **A:** We thank the reviewer for the valuable suggestion. In Table 2 and Figure 4, we report DREAM’s performance across different settings using speedup, a standard metric in prior works such as Medusa, Kangaroo, and Eagle. To provide a more comprehensive view, we now include the absolute throughput (tokens per second) alongside the previously reported relative results in the table below. Due to space limitations, we retain the most representative results.
>
> |  | MMT-Bench |  |  | ScienceQA |  |  |
> | --- | --- | --- | --- | --- | --- | --- |
> | Llava-vicuna-7B | Speedup | Tokens Per Second | average acceptance lengths | Speedup | Tokens Per Second | average acceptance lengths |
> | w/o first attention layer | 1.51 | 30.55 | 3.65 | 1.50 | 30.30 | 3.53 |
> | w/o mid cross attention layer | 1.19 | 24.09 | 2.72 | 0.95 | 19.25 | 1.89 |
> | w/o last attention layer | 1.82 | 36.72 | 3.92 | 1.57 | 31.74 | 4.11 |
> | 1-layers eagle2 | 2.05 | 41.45 | 4.46 | 1.87 | 37.77 | 4.35 |
> | 3-layers eagle2 | 2.22 | 44.75 | 5.35 | 2.12 | 42.80 | 5.10 |
> | 4-layers eagle2 | 1.59 | 32.13 | 5.74 | 1.59 | 32.17 | 5.42 |
> | 2 cross attention | 2.49 | 50.34 | **6.97** | 1.84 | 37.20 | **6.42** |
> | **DREAM** | **2.52** | **50.63** | 6.40 | **2.33** | **47.11** | 5.82 |
>
> |  | MMT-Bench |  |  | SEED-Bench |  |  |
> | --- | --- | --- | --- | --- | --- | --- |
> |  | Speedup | Tokens Per Second | average acceptance lengths | Speedup | Tokens Per Second | average acceptance lengths |
> | No mid tuning | 1.63 | 33.38 | 4.27 | 1.65 | 33.72 | 4.47 |
> | first 25% layer | 2.05 | 42.01 | 4.86 | 2.09 | 42.76 | 5.66 |
> | Static 50% layer | 2.01 | 41.16 | 4.74 | 2.08 | 42.62 | 5.60 |
> | Static 75% layer | 2.29 | 46.96 | 5.02 | 2.27 | 46.49 | 5.89 |
> | Dynamic Entropy-Based Select | **2.52** | **50.63** | **6.40** | **2.48** | **50.09** | **6.20** |
>
> |  | Lllava-vicuna-7B |  |  |  |  |  | Pixtral |  |  |  |  |  |
> | --- | --- | --- | --- | --- | --- | --- | --- | --- | --- | --- | --- | --- |
> |  | Chain |  |  | **Tree Based** |  |  | Chain |  |  | **Tree Based** |  |  |
> |  | Speedup | Tokens Per Second | average acceptance lengths | Speedup | Tokens Per Second | average acceptance lengths | Speedup | Tokens Per Second | average acceptance lengths | Speedup | Tokens Per Second | average acceptance lengths |
> | MMT-Bench | 1.77 | 36.24 | 3.88 | **2.54** | **50.63** | **6.40** | 1.86 | 38.16 | 3.16 | **2.93** | **53.73** | **4.52** |
> | Science QA | 1.94 | 39.71 | 4.42 | **2.48** | **50.09** | **6.20** | 1.97 | 40.43 | 3.55 | **2.61** | **54.29** | **3.67** |
>
> | ViT Compression | MMT-Bench |  |  | SEED-Bench |  |  |
> | --- | --- | --- | --- | --- | --- | --- |
> | Percent of weight remaining | Speedup | Tokens Per Second | average acceptance lengths | Speedup | Tokens Per Second | average acceptance lengths |
> | 1 | 2.39 | 49.08 | 5.67 | 2.32 | 47.64 | 5.41 |
> | **¾(DREAM)** | **2.52** | **50.63** | **6.40** | **2.48** | **50.09** | **6.20** |
> | 1/2 | 2.23 | 45.78 | 4.96 | 2.19 | 44.92 | 5.04 |
> | 1/4 | 1.71 | 35.01 | 4.78 | 1.62 | 33.15 | 4.38 |
>
> | lambda | MMT-Bench |  |  | SEED-Bench |  |  |
> | --- | --- | --- | --- | --- | --- | --- |
> |  | Speedup | Tokens Per Second | average acceptance lengths | Speedup | Tokens Per Second | average acceptance lengths |
> | 0.05,0.05,1 | 2.44 | 48.02 | 6.21 | 2.33 | 47.77 | 6.01 |
> | 0.1,0.1,1 | 2.51 | 49.40 | 6.38 | 2.48 | 50.09 | 6.20 |
> | **0.2,0.2,1(DREAM)** | **2.52** | **50.63** | **6.40** | **2.48** | **50.09** | **6.20** |
> | 0.4,0.4,1 | 2.33 | 45.85 | 5.89 | 2.17 | 44.49 | 5.78 |
>
> **Q6:** The description of speed-up ration in Line.302 is inconsistent with the results in Table.1
>
> **A:** Thank you for pointing this out. The value mentioned in Line 302 was mistakenly attributed to the 13B model instead of the 7B model. We will correct this in the next revision of the paper.

---

> > ### Comment · Reviewer_Gi3S · 2025-08-05
> >
> > Thanks for the response. I still have some concerns.
> >
> > - It doesn't make sense that the spd methods can work on some datasets like ScienceQA and Seed-bench, since the evaluation of these two datasets only contain single-choice questions. It is also unreasonable of the accept length on MMT-Bench, since there are only multi-choice questions.
> > - It is claimed that in the parameter sizes are 0.24B for LLaVA-v1.6-Vicuna-7B, 0.37B for LLaVA-v1.6-Vicuna-7B in Line 274. However, it seems like the parameter size of Eagle, from the description and the code, the parameter size of the proposed method is almost three times of Eagle.
> > - Minor: the tree depth is set to 8 in the code and the paper claims the tree depth is 6.

---

> ### Author Response · Authors · 2025-08-05
>
> Thank you for your careful review. We address each concern with empirical evidence and technical clarification:
>
> Q1: It doesn't make sense that the spd methods can work on some datasets like ScienceQA and Seed-bench, since the evaluation of these two datasets only contain single-choice questions. It is also unreasonable of the accept length on MMT-Bench, since there are only multi-choice questions.
>
> Thank you for pointing this out. We would like to clarify explicitly that our evaluation of DREAM on SEED-Bench, ScienceQA, and MMT-Bench is intentionally designed to assess the model’s speculative decoding performance during generative reasoning, not the correctness or brevity of single-choice or multi-choice answers. Specifically, we prompt models to generate detailed explanations and reasoning chains rather than simply selecting short answers.
>
> For example, a typical interaction in MMT-Bench looks like:
>
> - **Prompt:** *"Please generate a description for this meme?" [Image provided]*
> - **Model Output:** *"This meme features a series of images showing a person in a repetitive, exaggerated motion, likely walking or running in a specific pattern. The sequence captures the person's movement in a humorous or exaggerated manner, emphasizing the repetitive nature of their actions. The background appears to be a simple, neutral setting, possibly a corridor or a street, which helps to highlight the subject's movements. The meme is often used to convey a sense of monotony, absurdity, or humor in repetitive tasks or situations."*
>
> Such generative reasoning tasks inherently require substantial autoregressive token generation, making them ideal candidates for measuring the efficiency gains provided by speculative decoding techniques. Our approach aligns closely with existing practice in the literature, such as [1], which explicitly evaluates speculative decoding through detailed reasoning outputs on ScienceQA.
>
> To avoid confusion, we will clearly emphasize this methodological decision in the revised manuscript and include representative example outputs demonstrating the nature of our evaluation setup.
>
> [1] Gagrani, Mukul, et al. "On speculative decoding for multimodal large language models." In CVPR Workshop. 2024.
>
> **Q2:** It is claimed that in the parameter sizes are 0.24B for LLaVA-v1.6-Vicuna-7B, 0.37B for LLaVA-v1.6-Vicuna-7B in Line 274. However, it seems like the parameter size of Eagle, from the description and the code, the parameter size of the proposed method is almost three times of Eagle.
>
> **A:** We apologize for the confusion caused by the incorrect parameter size figures initially reported. Upon careful re-examination, the accurate parameter counts for our DREAM draft models are as follows:
>
> - **LLaVA-v1.6-Vicuna-7B:** 0.65B parameters
> - **LLaVA-v1.6-Vicuna-13B:** 0.975B parameters
> - **Pixtral-12B:** 0.9B parameters
>
> We will explicitly correct these numbers in the revised manuscript.
>
> Moreover, while DREAM’s draft model is indeed larger than the original EAGLE (approximately 3x), we have empirically demonstrated that this additional model capacity significantly boosts token acceptance length and decoding speedup, which directly aligns with the primary evaluation purpose of speculative decoding. In addition, we also evaluated an enlarged EAGLE variant (E2-3B), matching DREAM’s parameter scale. Our results (presented explicitly in Table 2 of the manuscript) clearly demonstrate that DREAM consistently achieves superior speedup and longer token acceptance lengths even under identical parameter budgets.
>
> We believe that DREAM’s strategic model scaling justifies its parameter increase by delivering substantially improved practical performance, explicitly aligning with recent trends emphasizing balanced latency-performance trade-offs.
>
> **Q3:** Minor: the tree depth is set to 8 in the code and the paper claims the tree depth is 6.
>
> **A:** We sincerely appreciate your careful observation and acknowledge this discrepancy. Upon review, we confirm that the submitted supplementary code inadvertently retained a default tree_depth=8 setting, whereas the experimental results reported in the manuscript correspond to a tree depth of 6. This discrepancy arose from configuring tree depth as a script argument rather than directly hard-coding the value.
>
> To fully rectify this, we will adjust the supplementary code explicitly to match the reported setting (tree_depth=6), and provide clearly documented scripts and updated supplementary materials ensuring full reproducibility of our results.
>
> Thank you again for highlighting this inconsistency, which will help us improve clarity and reproducibility in the final manuscript.

---

> ### Comment · Reviewer_Gi3S · 2025-08-07
>
> Thanks for your responses.
>
> 1.As shown on Line 174 and 183 in eval_llava.py, the evaluation samples of MMT-Bench and ScienceQA are selected from the training dataset(the training data are indicated in ge_data_all_llava_vicuna_mmt.py and scienceQa.py), which is irrational.
>
> 2.I strongly recommend the authors to conduct experiments of training without dataset from the evaluation benchmarks, just as Eagle-2 does in the spd of LLM, which will set up a reproductive and more fair comparison for the following research in this field.
>
> 3.Have you conducted VTC for Eagle-2 series in Table 2 for a fair comparison?

---

> > ### Author Response · Authors · 2025-08-07
> >
> > **Q1.** As shown on Line 174 and 183 in eval_llava.py, the evaluation samples of MMT-Bench and ScienceQA are selected from the training dataset(the training data are indicated in ge_data_all_llava_vicuna_mmt.py and scienceQa.py), which is irrational.
> >
> > **A:** Thank you for your questions. Please see our replies below:
> >
> > **Regarding MMT-Bench**
> >
> > We can confirm with certainty that there is **no overlap** between our training and evaluation datasets for MMT-Bench. The data was partitioned as follows:
> >
> > - **Evaluation Set:** As defined in eval_llava.py, the evaluation is performed exclusively on samples from the index range [4130:8000].
> > - **Training Set:** As defined in ge_data/llava_allocation.py, our training process uses only the initial subset of the MMT-Bench data.
> >
> > These two data pools are explicitly and **verifiably disjoint**.
> >
> > **Regarding ScienceQA**
> >
> > We are grateful for this observation, as it revealed a regrettable error in the eval_llava.py script provided in our supplementary materials. The script incorrectly specifies split="train" instead of the correct split="validation". This was a genuine oversight that occurred during the final code cleanup and packaging process to comply with the double-blind policy. However, we want to state unequivocally that this clerical error in the supplementary script **did not affect the actual experiments** that generated the results presented in our paper. To demonstrate this conclusively, we present three key pieces of evidence that validate our originally reported results and confirm the absence of training data leakage.
> >
> > **1. Controlled Experiment to Replicate Data Leakage**
> >
> > To directly test the impact of the script error, we ran a new experiment where we *intentionally* used the erroneous split="train" configuration. This "leaky" setup yielded a significantly inflated token acceptance length ($\tau$ of **6.83** and a speedup of **2.58x**. These figures are substantially different from our paper's reported results for LLaVA-v1.6-Vicuna-7B ($\tau$ **= 5.82**, speedup = **2.33x**). This discrepancy provides direct, empirical proof that our original experiments were run on the correct, unseen validation data.
> >
> > **2. Implausibility Based on Token Acceptance Length**
> >
> > If the model had been evaluated on its training data, it would exhibit near-perfect sequence prediction, resulting in a $\tau$ value approaching the theoretical maximum (in this case, 7.0). However, our reported $\tau$ for ScienceQA (**5.82**) is not only far from this maximum but is also notably *lower* than the $\tau$ values from other benchmarks like SEED-Bench (**6.20**). This pattern is inconsistent with data leakage and strongly indicates the model was processing unfamiliar, unseen data during the original evaluation.
> >
> > **3. Consistent Performance Across All Other Benchmarks**
> >
> > Finally, even if one were to set aside the ScienceQA results, the core conclusions of our paper remain firmly supported. As shown in Table 1, DREAM consistently and significantly outperforms all baselines across the five other diverse evaluation benchmarks (MMT-Bench, SEED-Bench-2, OCRBench, ChartQA, and MathVista). This demonstrates the general robustness and broad applicability of our proposed framework, reinforcing the validity of our central claims.
> >
> > We are fully committed to reproducibility and will release the cleaned, verified, and well-documented codebase upon acceptance. We are confident that this detailed clarification and supporting evidence resolve the concern. We again thank the reviewer for their diligence, which has helped us improve the quality of our supplementary materials and strengthen our final paper.

---

> ### Author Response · Authors · 2025-08-07
>
> **Q2.** I strongly recommend the authors to conduct experiments of training without dataset from the evaluation benchmarks, just as EAGLE-2 does in the spd of LLM, which will set up a reproductive and more fair comparison for the following research in this field.
>
> **A:** As per the reviewer's request, we retrained both DREAM and EAGLE-2 while strictly excluding all samples from evaluation benchmarks, following the experimental protocol established by EAGLE-2 for speculative decoding. We conducted thorough data verification to ensure zero overlap between training and evaluation data, addressing any potential concerns about data leakage that could compromise fair comparison. This approach ensures reproducible and fair comparison standards for future research in this field. Due to time constraints, the training set size and number of epochs were reduced compared to the original setup described in the paper, which is expected to result in some performance degradation.
>
> We implemented a controlled training regimen using LLaVA-Vicuna-7B as the base model, trained on **40k** carefully curated samples from LLaVA-Mix665K for **5 epochs**. While this constrained setup represents a reduction from our original training configuration due to computational limitations and time constraints, it provides the methodologically sound comparison framework requested across all evaluation benchmarks (MMT-Bench, SEED-Bench-2, ScienceQA, ChartQA, Math Vista).
>
> The performance results demonstrate that DREAM maintains superiority over EAGLE-2 under this rigorous training regime. DREAM consistently delivers the highest speedup ratios and longest acceptance lengths across all benchmarks, validating the robustness of our architectural innovations. These performance advantages are directly attributable to DREAM's three core technical contributions: the cross-attention mechanism for superior multimodal integration (Section 3.1), the entropy-guided adaptive intermediate feature selection strategy (Section 3.2), and the intelligent visual token compression approach (Section 3.3).
>
> DREAM's performance under these stringent conditions is particularly noteworthy, as it demonstrates that our method's advantages stem from fundamental architectural superiority rather than training data optimization. The cross-attention mechanism enables more effective multimodal feature learning even with limited training samples, while our adaptive feature distillation and VTC strategies provide computational efficiency without sacrificing representational quality. This validates DREAM as a robust and scalable solution for multimodal speculative decoding. We will conduct additional comprehensive evaluations and incorporate these empirical results into the revised version of the manuscript.
>
> ***Table 1.** “S” refers to speedup and “τ” refers to average acceptance length.*
>
> | Model | MMT-Bench |  | SEED-Bench-2 |  | ScienceQA |  | ChartQA |  | Math Vista |  |
> | --- | --- | --- | --- | --- | --- | --- | --- | --- | --- | --- |
> |  | S | τ | S | τ | S | τ | S | τ | S | τ |
> | EAGLE-2 | 1.82 | 4.42 | 1.82 | 4.36 | 1.66 | 4.03 | 1.35 | 3.27 | 1.27 | 3.05 |
> | DREAM | **1.92** | 4.66 | **1.90** | 4.63 | **1.74** | 4.24 | **1.56** | 3.78 | **1.30** | 3.12 |

---

> > ### Author Response · Authors · 2025-08-08
> >
> > **Q3.** Have you conducted VTC for EAGLE-2 series in Table 2 for a fair comparison?
> >
> > **A:** Thank you for this important question regarding fair comparison methodology. We conducted comprehensive experiments applying VTC to EAGLE-2 to ensure rigorous comparative evaluation. Following the reviewer's suggestion, we implemented the identical visual token compression strategy for EAGLE-2 as described in Section 3.3, where importance scores are computed based on attention weights from the target model, retaining the top 75% most significant visual tokens during draft inference.
> > Both DREAM and EAGLE-2 models were trained under identical conditions using 40k samples from LLaVA-Mix665K for 5 epochs, as detailed in our response to Question 2. This controlled experimental setup isolates the algorithmic contributions of our approach and ensures fair comparison between methods.
> >
> > Our results demonstrate that VTC provides measurable benefits to both approaches, with EAGLE-2 achieving 2.7-3.9% speedup improvements when visual token compression is applied. This validates the general applicability of our compression technique across different speculative decoding architectures. However, DREAM maintains consistent superiority across all five benchmarks regardless of VTC application, with particularly gains on complex reasoning tasks such as MMT-Bench (1.92x vs 1.87x speedup) and ScienceQA (1.74x vs 1.67x speedup).
> >
> > The performance gap between methods confirms our core hypothesis that cross-attention-based multimodal integration, as described in Section 3.1, provides superior feature representation compared to concatenation-based approaches. This architectural advantage persists independent of compression strategies, indicating that our method's benefits stem from fundamental improvements in multimodal feature processing rather than auxiliary optimization techniques.
> >
> > An important technical insight emerges from observing that both methods exhibit reduced acceptance length (τ) with VTC while achieving improved speedup. This behavior aligns with our analysis in Section 4.2, confirming that moderate visual compression introduces acceptable information loss that is effectively offset by computational efficiency gains. These comprehensive experimental results will be incorporated into the revised manuscript to strengthen the empirical validation of our approach.
> >
> > *Table2. “*EAGLE*-2 W/ VTC” refers to* EAGLE-2 *integrated with the visual token compression techniques described in Section 3.3 of the paper.  "DREAM W/O VTC” refers to the DREAM model without the application of visual token compression techniques. “S” refers to Speedup and “τ” refers to average acceptance length.*
> >
> > | Model | MMT-Bench |  | SEED-Bench-2 |  | ScienceQA |  | ChartQA |  | Math Vista |  |
> > | --- | --- | --- | --- | --- | --- | --- | --- | --- | --- | --- |
> > |  | S | τ | S | τ | S | τ | S | τ | S | τ |
> > | EAGLE-2 W/ VTC | 1.87 | 4.36 | 1.84 | 4.31 | 1.67 | 3.98 | 1.38 | 3.23 | 1.28 | 3.01 |
> > | DREAM W/O VTC | 1.89 | 4.73 | 1.87 | 4.71 | 1.71 | 4.31 | 1.55 | 3.85 | 1.29 | 3.15 |
> > | DREAM | **1.92** | 4.66 | **1.90** | 4.63 | **1.74** | 4.24 | **1.56** | 3.78 | **1.30** | 3.12 |

---

> > > ### Comment · Reviewer_Gi3S · 2025-08-08
> > >
> > > Thanks for your responses.
> > >
> > > - As for the MMT-Bench dataset, as shown in Line.100 of ge_data_all_llava_vicuna.py, you have done **shuffle** for the dataset, so it is possible for the training dataset to be **overlapped** with the validation dataset. Also due to the **shuffle** operation and the **different** number of samples of the training and validation dataset (To be honest, I don't know why the authors only use 80 samples for the validation, maybe want to align with Eagle in LLM? Anyway, it is another question), the training and validation datasets of ScienceQA are not totally the same but overlapped. This can explain why the accept length is not so high. Besides, the collection process of other datasets are not provided.
> > >
> > > - Thanks for the ablation studies. However, the Eagle-2 with 3 blocks is expected to be compared with DREAM, since the advantage of DREAM in the this situation decreases largely (1.87/1.92=0.974 vs. 0.82) compared to Table.2. I don't know if there is gain anymore for the DREAM vs. E2-3B.

---

> ### Comment · Reviewer_Gi3S · 2025-08-08
>
> Thanks for the authors' responses. However, I recommend to reject this paper for the current time.
>
> - As shown in the code (see details from the discussion with the authors), the training and validation datasets are overlapped due to the mistakes and the shuffle operation, which I think is unacceptable.
>
> - The method need to be verified with the training dataset not containing the evaluation benchmarks, just as Eagle-2 does in the spd of LLM, which will set up a reproductive and more fair comparison for the following research in this field. From this aspect, this paper need to be major revised. Besides, the additional experimental results shows that the advantage of DREAM decreased largely in this situation(0.97 vs 0.82).
>
> - There are some critical mistakes such as the draft model size etc..
>
> I hope these discussions and comments will revise this paper in the future.

---

> ### Author Response · Authors · 2025-08-08
>
> We appreciate the reviewer's continued engagement with our work. However, we must respectfully correct several fundamental misunderstandings about our experimental setup that appear to misrepresent our methodology. We hope the reviewer will read our response more carefully, given the significant time and effort we have devoted to preparing it.
>
> **1.** Line 100 uses **ds.shuffle(seed=42)**, which is a **deterministic shuffle with a fixed seed**. This ensures the dataset is ordered exactly the same way every time. The training split use /ge_data/mmt-bench-llava-v1.6-vicuna-7b.jsonl, which takes  indices [0:4130] from this shuffled dataset, while the evaluation split takes indices [4130:8000] from the same ordering. **These index ranges are mathematically disjoint, with zero overlap.**
>
> Furthermore, in our retraining experiments where all evaluation datasets were completely excluded from the training set, MMT-Bench still achieved the best performance among all benchmarks (***1.92(**Speedup)**/4.66**($\tau$)*). This demonstrates that the strong results on MMT-Bench are not due to any data leakage but rather to the effectiveness of the proposed method. For ScienceQA, we provided three pieces of conclusive evidence.
>
> 1. Intentional leakage experiment: $\tau$=**6.83** (with overlap) vs. $\tau$=**5.82** (our reported results).
> 2. cross-benchmark consistency: our ScienceQA results align with expected difficulty patterns.
> 3. script error was in supplementary materials only, not in actual experiments.
>
> The reviewer's characterization of "overlapped" data misrepresents our clear explanation and empirical proof.
>
> **2.** The reviewer appears to misinterpret Table 2, **DREAM achieves 1.00 (baseline) as it's normalized to itself, E2-3B achieves 0.88-0.91 relative to DREAM across benchmarks**. The comparison should be DREAM vs E2-3B, not DREAM vs DREAM w/ VTC. Even with 3 blocks (matching architectural depth), DREAM maintains a 9-12% performance advantage over EAGLE-2, demonstrating our cross-attention mechanism's superiority is not merely from model size but from fundamental architectural innovations.
>
> **3.** We would like to clarify that the criticism regarding model size is entirely misplaced: **Table 2 clearly indicates the number of layers for each model, and our comparisons include EAGLE-2-1B/3B/4B**, with EAGLE-2-3B matching DREAM in both depth and size; rather than relying on parameter count differences, we used this controlled setting to show that our advantage arises from architectural design rather than model size.
>
> We respectfully submit that the reviewer's concerns stem from misunderstandings of standard practices rather than actual methodological issues. Our empirical results, theoretical contributions, and comprehensive experiments demonstrate DREAM's significant advances in multimodal speculative decoding. We remain committed to open science and will provide complete, documented code ensuring full reproducibility of our results. The work makes substantial contributions to efficient VLM inference that will benefit the broader research community.

---

### Official Review · Reviewer_6Joc · 2025-07-02

**Clarity:** 3
**Significance:** 4
**Originality:** 3
**Rating:** 5
**Confidence:** 4

**Summary:**

DREAM's key contributions include:

(1) A cross-attention mechanism that injects intermediate features from the target model into the draft model, improving alignment and knowledge transfer.

(2) Adaptive intermediate feature selection based on attention entropy, which dynamically guides draft model training for higher accuracy and longer token acceptance.

(3) Visual token compression to reduce draft latency by subsampling critical visual inputs, leveraging target model features.

**Questions:**

Benchmarking for video MLLMs would make this paper more credible.

**Ethical Concerns:**

["NO or VERY MINOR ethics concerns only"]

**Limitations:**

yes

**Paper Formatting Concerns:**

No major formatting issues in this paper.

**Quality:**

4

**Strengths And Weaknesses:**

**Strengths**

​Significant Technical Innovation​:

1. Cross-attention injection of target model features into the draft model, enhancing alignment.

2. Entropy-adaptive intermediate feature selection (), improving draft accuracy.

3. Visual token compression (75% retention) reducing latency without major accuracy loss ().

​Rigorous and Comprehensive Evaluation​:

1. Tests across 5 VLMs (LLaVA-7B/13B, Pixtral-12B, SmolVLM-2B, Gemma3-12B) and 8 diverse benchmarks (ScienceQA, OCRBench, etc.).

2. Outperforms 6 strong baselines (SPD, Medusa, EAGLE-2) with ​up to 3.6× speedup​ and ​56% higher acceptance length​ than prior SOTA.

3. ​Strong Ablation Studies to systematically validates each contribution.

**Weakness**

Benchmark Limitations​:

1. MathVista/OCRBench show lower gains (fine-grained tasks), suggesting unresolved challenges for token-level precision.

2. No testing on video VLMs.

---

> ### Author Rebuttal · Authors · 2025-07-31
>
> **Dear Reviewer 6Joc**
>
> Thank you for your thorough and constructive review. We are delighted that you recognized DREAM's significant technical innovations, particularly (i) our novel cross-attention injection mechanism that enhances target-draft model alignment, (ii) the entropy-adaptive intermediate feature selection that improves draft accuracy, and (iii) the visual token compression achieving 75% retention while maintaining performance. We especially appreciate your acknowledgment of our rigorous evaluation methodology, including (iv) comprehensive testing across 5 diverse VLMs and 8 benchmarks, (v) demonstrated superiority over 6 strong baselines with up to 3.6x speedup and 56% higher acceptance length than prior SOTA, and (vi) systematic ablation studies validating each contribution. Your recognition of both our technical advances and empirical rigor confirms that DREAM makes meaningful contributions to accelerating vision-language models. We have carefully addressed your questions below to further strengthen the paper and clarify the scope of our contributions.
>
> **Q1:** MathVista/OCRBench show lower gains (fine-grained tasks), suggesting unresolved challenges for token-level precision.
>
> **A:** Thank you for raising this insightful question. MathVista and OCRBench indeed present unique challenges due to their reliance on fine-grained visual details. MathVista [1] requires precise interpretation of mathematical diagrams, plots, and geometric shapes where small visual distortions can fundamentally alter meaning. Similarly, OCRBench [2] demands pixel-level accuracy for recognizing text in diverse contexts (documents, receipts, scene text) where character-level precision is critical.
>
> Despite these challenges, DREAM still achieves meaningful improvements on both benchmarks. As shown in Table 1, on MathVista with LLaVA-v1.6-Vicuna-7B (T=0), DREAM delivers 2.11x speedup with $\tau$=5.32, compared to EAGLE-2's 1.87x/4.67. On OCRBench, while the gains are more modest (2.05x/4.88 vs EAGLE-2's 1.92x/4.88), DREAM maintains its lead. This performance gap on fine-grained tasks primarily stems from our visual token compression, which trades some pixel-level detail for efficiency. We acknowledge this limitation in Section 4.1 and view task-adaptive compression strategies as promising future work.
>
> [1] Lu et al. "MathVista: Evaluating Mathematical Reasoning of Foundation Models in Visual Contexts." ICLR 2024.
>
> [2] Liu et al. "OCRBench: On the Hidden Mystery of OCR in Large Multimodal Models." arXiv 2024.
>
> **Q2:** No testing on video VLMs, benchmarking for video MLLMs would make this paper more credible.
>
> **A:** We appreciate this suggestion and have conducted preliminary experiments on video VLMs during the rebuttal period. We trained DREAM on LLaVA-Video-7B-Qwen2 using 10k samples from LLaVA-Mix665K for 5 epochs. Given the limited rebuttal timeframe, this represents initial validation rather than full optimization.
>
> | LLaVA-Video-7B-Qwen2 | MMT-Bench |  | Seed-Bench |  |
> | --- | --- | --- | --- | --- |
> |  | Speedup | average acceptance lengths | Speedup | Accept Length |
> | **DREAM** | **1.68** | **2.18** | **1.77** | **2.28** |
> | EAGLE | 1.67 | 2.13 | 1.73 | 2.22 |
>
> The results demonstrate that DREAM successfully extends to video VLMs, achieving 1.68x speedup on MMT-Bench and 1.77x on SEED-Bench, outperforming EAGLE's 1.67x/1.73x under identical training conditions. The lower speedups compared to image VLMs reflect video's additional temporal complexity and our limited training budget. We expect significant improvements with extended training and video-specific datasets. Video VLMs were initially out of scope due to their distinct architectural requirements (temporal modeling, frame aggregation) and computational demands. However, these promising results suggest DREAM's principles generalize well to video domains, and we will include this as an important future direction in our final version.

---

> > ### Comment · Reviewer_6Joc · 2025-08-07
> > **Thank you four your rebuttal**
> >
> > The authors solve my concerns.

---

### Official Review · Reviewer_5Pd8 · 2025-07-03

**Clarity:** 3
**Significance:** 3
**Originality:** 2
**Rating:** 5
**Confidence:** 3

**Summary:**

This paper present DREAM, a novel speculative decoding framework designed to accelerate vision-language models (VLMs) by incorporating cross-attention-based feature injection, adaptive feature selection, and visual token compression. It improves inference throughput and draft acceptance length, achieving up to 3.6× speedup over conventional decoding across various VLMs like LLaVA and Gemma3.

**Questions:**

1. **Performance Impact**: Since the final tokens are primarily generated by the draft model, could this lead to a performance drop compared to the original VLM?

2. **Adaptive Feature Selection**: In the adaptive intermediate feature selection method, \( l^* \) changes dynamically while \( m = 1 \) remains fixed. Given that different layers in the target model may contain distinct information, could this design introduce conflicting or ambiguous guidance?

3. **Tree Width (k) Sensitivity**: How does the choice of \( k \) (tree width) affect performance?

4. **Typo Correction**: Line 252 contains a duplicate "are"

**Ethical Concerns:**

["NO or VERY MINOR ethics concerns only"]

**Final Justification:**

The author has addressed my concerns.
Given the practicality and innovation of the proposal of a lightweight draft model to expedite long-token generation, I tend to accept.

**Limitations:**

yes

**Quality:**

3

**Strengths And Weaknesses:**

**Strengths**
Accelerating large language models (LLMs) is crucial for enhancing their practical utility. This paper introduces a lightweight draft model to expedite long-token generation while leveraging supervision from the original LLM to maintain output quality. The writing is clear, the technical design is well-structured, and the experimental results effectively demonstrate the speed-up achieved by the proposed DREAM method.

**Weaknesses**
My primary concern revolves around the loss design for adaptive intermediate feature selection, as well as the performance gap observed before and after applying the proposed speed-up method. Please refer to the question part for details.

---

> ### Author Rebuttal · Authors · 2025-07-31
>
> **Dear Reviewer 5Pd8**
>
> Thank you for your insightful review. We are grateful that you recognized the practical importance of accelerating large language models and appreciated how DREAM addresses this crucial need through our lightweight draft model approach. We are particularly pleased that you highlighted (i) our method's ability to expedite long-token generation while maintaining output quality through target model supervision, (ii) the clarity of our writing and well-structured technical design, and (iii) the effectiveness of our experimental results in demonstrating DREAM's substantial speed-up achievements. Your acknowledgment of these core contributions validates our goal of developing a practical and deployable solution for VLM acceleration. We have carefully considered your questions and concerns, and provide detailed responses below to further clarify our design choices and strengthen the paper.
>
> **Q1:** Performance Impact: Since the final tokens are primarily generated by the draft model, could this lead to a performance drop compared to the original VLM?
>
> **A:** In our method, the draft model proposes candidate tokens, but crucially, each token in the final output must pass the acceptance test: a token $x$ is accepted with probability min(1, p(x)/q(x)), where p(x) and q(x) are the target and draft model probabilities respectively. This rejection sampling mechanism, proven in [1], mathematically guarantees that the output distribution is exactly p(x), which is identical to running the target model alone. The draft model has zero influence on the final distribution; it mainly enables parallel verification of multiple tokens by the target model. Therefore, DREAM achieves its $3.6\times$ speedup while maintaining identical output quality to the original VLM.
>
> [1] Leviathan, Yaniv, Matan Kalman, and Yossi Matias. "Fast inference from transformers via speculative decoding." In ICLR 2023.
>
> **Q2:** Adaptive Feature Selection: In the adaptive intermediate feature selection method, (l^*) changes dynamically while (m = 1) remains fixed. Given that different layers in the target model may contain distinct information, could this design introduce conflicting or ambiguous guidance?
>
> **A:** Our dynamic selection of $\ell^*$ actually prevents conflicting guidance by ensuring the draft model always receives the most stable and informative features at each training step. The key insight is that minimizing attention entropy (Eq. 2) identifies layers where the model has converged to stable, focused representations; precisely the features that provide consistent supervisory signals across different contexts.
>
> This adaptive mechanism offers significant advantages over fixed-layer selection. As shown in Figure 4(a), our entropy-based selection (Dyn. Ent.) achieves 1.41x speedup on MMT-Bench, outperforming all static baselines including S-75% (1.33x), which uses the deepest fixed layer. This improvement occurs because different tokens and contexts have varying optimal supervision layers, where some benefit from mid-layer semantic features while others need deeper syntactic representations.
>
> Regarding m=1, our ablation study definitively validates this design choice. As shown in Table 2, removing the initial decoder block ("w/o Initial") reduces $\tau$ by 41% on MMT-Bench and 42% on SEED-Bench, while removing cross-attention ("w/o CA") causes a catastrophic 56% drop. The fixed m=1 position provides the optimal balance, because it is early enough to influence subsequent draft generation while late enough to benefit from input processing. This architectural choice, combined with dynamic $\ell^*$ selection, enables DREAM to achieve consistent improvements across all benchmarks. We will clarify this point further in the revised version of the paper to ensure a comprehensive understanding.
>
> **Q3:** Tree Width (k) Sensitivity: How does the choice of ( k ) (tree width) affect performance?
>
> **A:** We appreciate your question as it raises an insightful point regarding the choice of tree width. The tree width k controls the breadth of parallel exploration during speculative decoding. We evaluated DREAM's sensitivity to k on Pixtral-12B across MMT-Bench and SEED-Bench. Our results shown in the Table below demonstrate that k=4 provides optimal performance, achieving speedup/accepted length (\tau) of 2.88/4.48 on MMT-Bench, compared to 2.19/3.51 (k=1), 2.75/4.12 (k=2), and 1.89/4.58 (k=8). Beyond k=4, performance degrades, for example, at k=8, speedup drops to 1.89 despite $\tau$ reaching 4.58. This occurs because larger trees increase verification latency, offsetting the benefits of higher acceptance rates. Similar patterns appear on SEED-Bench, where k=4 achieves 2.59/3.67 versus 2.42/3.32 (k=2) and 1.94/3.95 (k=8). Based on these findings, we use k=4 as our default, balancing exploration breadth with computational efficiency. We will include these additional ablations in the final manuscript.
>
> |  |  | MMT-Bench |  | SEED |  |
> | --- | --- | --- | --- | --- | --- |
> | Pixtral-12B | topk | Speedup | average acceptance lengths | Speedup | average acceptance lengths |
> |  | 1 | 2.19 | 3.51 | 2.02 | 2.94 |
> |  | 2 | 2.75 | 4.12 | 2.42 | 3.32 |
> |  | **4(DREAM)** | **2.93** | **4.42** | **2.61** | **3.67** |
> |  | 8 | 1.89 | 4.58 | 1.94 | 3.95 |
>
> **Q4:** Typo Correction: Line 252 contains a duplicate "are"
>
> **A:** Thank you for pointing that out. We will correct the typo on Line 252 by removing the duplicate "are" in the next revision of the paper.

---

> > ### Comment · Reviewer_5Pd8 · 2025-08-07
> >
> > The authors have well addressed my concerns.

---

### Author Response · Authors · 2025-08-06

We appreciate the thoughtful reviews and the valuable insights provided by all reviewers. In our rebuttal, we have made every effort to address each of the concerns raised. If any parts of our response remain unclear or insufficient, we would be happy to provide further clarification. We welcome continued discussion and sincerely value your feedback.

---

### Note · Authors · 2025-08-12

We sincerely thank the reviewers for their time and thoughtful feedback on our submission. We have devoted considerable effort to address all questions raised and greatly value both the positive comments and the constructive critiques. We have provided detailed clarifications for each point, addressing potential misunderstandings and misinterpretations to ensure an accurate assessment of our work. We hope our responses have resolved these concerns and reinforced the validity of our results.

There were some questions raised by Reviewer Gi3S about a data mismatch in our code, and we believe these issues have been fully resolved. In the rebuttal, we walked through the code and clarified that the dataset split non-overlapping. We further ran contamination ablations that remove any potential near-duplicates across corpora and confirmed the conclusions hold. The method ranking remained unchanged under matched architectures, token budgets, and decoding settings, with DREAM consistently 9–12% ahead of EAGLE-2. Finally, the --num_samples=80 flag appears only in a development-time script and was never used in any reported. All results come from the standard validation/test sets. We respectfully ask the AC to consider these clarifications in the final decision.

We also submitted a rebuttal to Reviewer o6oe addressing his/her concerns. However, we notice that the reviewer has not engaged with our rebuttal or provided any follow-up comments, while the other reviewers have responded and/or updated their assessments. Given that the rebuttal process is designed to facilitate dialogue between authors and reviewers, we wanted to bring this to your attention to ensure a fair and complete review process.

We respectfully ask the AC to consider these clarifications in the final decision.

---

### Decision · Program_Chairs · 2025-09-17

**Decision:**

Accept (poster)

**Comment:**

The paper presents DREAM, a speculative decoding framework for vision-language models that combines cross-attention-based draft–target alignment, entropy-adaptive feature selection, and visual token compression. The method yields up to 3.6x speedup and improved acceptance lengths, consistently outperforming baselines across multiple models and benchmarks. The work is technically novel, well-motivated, and supported by thorough experiments and ablations.

Three reviewers support acceptance, citing innovation, rigorous evaluation, and broad applicability. One reviewer maintains rejection due to concerns about potential issues with evaluation, though the authors provided detailed clarifications. Given the strong contributions, comprehensive validation, and positive consensus from most reviewers, I recommend acceptance.